# A Closer Look at Personalized Fine-Tuning in Heterogeneous Federated Learning

## Abstract

Federated Learning (FL) enables privacy-preserving, decentralized model training but faces significant challenges in balancing global generalization and local personalization due to non-identical data distributions across clients. While Personalized Fine-Tuning (PFT) adapts models to local data, excessive personalization often degrades global performance. In this work, we present a comprehensive empirical study encompassing seven diverse datasets, multiple model architectures, and various fine-tuning methods under both covariate and concept shift scenarios. Our extensive evaluation reveals critical limitations in existing PFT methods, which struggle with overfitting and exhibit inconsistent performance across distribution shifts, even with careful hyperparameter tuning and regularization. To address these issues, we identify LP-FT, a simple yet effective strategy that combines Linear Probing with full Fine-Tuning, adapted to the FL setting. LP-FT consistently outperforms existing methods, achieving an optimal balance between local personalization and global generalization across all tested scenarios. By investigating the feature change after PFT, we hypothesize the a phenomena dubbed as federated feature distortion is linked to the global generalization. Motivated by the observation, we provide a theoretical analysis of two-layer linear networks, offering novel insights into the conditions under which LP-FT excels, thereby enhancing our understanding of personalization dynamics in FL. This work contributes in three key areas: (1) a rigorous and comprehensive evaluation of PFT methods under diverse distribution shifts, (2) the introduction of LP-FT as a robust and versatile solution to FL personalization challenges, and (3) theoretical foundations that explain LP-FT's superior effectiveness. Our findings set a new venue for PFT research and provide valuable insights to the broader FL community.

## 1 Introduction

Federated Learning (FL) (McMahan et al., 2017) has emerged as a promising paradigm for collaborative learning from decentralized data, enabling multiple clients to train a shared global model, known as General FL (GFL), without compromising data privacy. A significant challenge in FL is the differing data distributions among clients, which undermines the effectiveness of conventional FL methods on individual clients.

In contrast to GFL, Personalized FL (PFL) (Kairouz et al., 2021) emerges as an approach to address this issue by customizing global models for each client. Clients can enhance the global model by applying local updates using their own datasets, a process known as *Personalized Fine-Tuning* (PFT) (Wu et al., 2022), as shown in Fig. 1 (a). However, this approach often causes models to overfit on local data, thereby compromising the benefits of FL and the generalization performance of the resulting FL models. This is particularly concerning in critical real-world applications, such as FL across multiple hospitals for disease diagnosis, where a local model must not only perform well on hospital patient data, but also generalize effectively to diverse patient populations that may be encountered on-site in the future (Xu et al., 2021). Therefore, balancing the optimization of individual client performance (personalization) with strong global performance (generalization across all or unseen clients) is crucial (Wu et al., 2022; Huang et al., 2024).

To address the personalization and generalization trade-off in PFT, a simple and commonly used strategy is personalized regularized fine-tuning, where each client fine-tunes a global model using

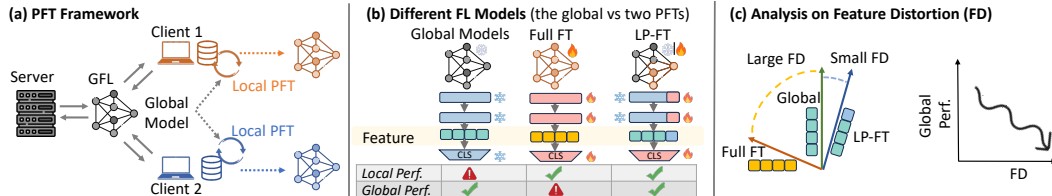

Figure 1: Overview of the problem setting and FL strategies investigated in this paper. (a) PFT framework, where each client fine-tunes a global model previously trained via GFL. (b) Three different PFT models: the global FL model, the full FT model, and the LP-FT model; their parameter updating patterns and local/global performance (perf.) under data heterogeneity; The fire icon indicates the actively tuned parameter, the frozen icon represents the fixed weight, and the mixed fire-frozen icon denotes the weight that is not actively tuned.). (c) Visualization of feature distortion under PFL and its possible link to global generalization.

their local data, with no further communication with the server. This method is appealing for its simplicity and broad applicability across datasets (Kumar et al., 2022), eliminating the need for complex model modifications or extensive data preprocessing. **The final local fine-tuning phase plays a crucial role**, as it influences both the model's adaptability to individual client data and its overall generalizability. Effective PFT can improve local performance without compromising global performance, whereas poorly executed PFT can result in severe overfitting (Wu et al., 2022).

In this work, we conduct comprehensive evaluation of various regularization-based PFT methods in heterogeneous FL environments under different distribution shift scenarios, categorized as covariate shift (Peng et al., 2019a; Hendrycks & Dietterich, 2019) and concept shift (Izmailov et al., 2022). Covariate and concept shifts are more prevalent in real-world scenarios where client data often varies in feature distributions or task definitions (Hendrycks & Dietterich, 2019). Despite meticulously tuning the hyper-parameters in conventional regularized fine-tuning methods, we observe persistent issues of local overfitting, wherein localized performance gains are achieved at the significant cost of compromised global generalization.

By observing this, we identify a straightforward alternative strategy—combining linear probing (LP) followed by full fine-tuning (FT), referred to as LP-FT (Kumar et al., 2022)—that consistently balances personalization and generalization across diverse distribution shifts, as shown in Fig. 1 (b). We hypothesize that LP-FT achieves this by preserving the representations learned in the last layer of the neural network (before the linear head) during local adaptation, thereby reducing the risk of *federated feature distortion*, as shown in Fig 1 (c). This phenomenon occurs when local fine-tuning disrupts these last-layer representations (features), which are critical for global performance, leading to performance degradation across the global client set. Although feature distortion was introduced in standard LP-FT approaches within the centralized domain (Kumar et al., 2022), the setting differs significantly from the one considered here.

To further support our findings on the superior performance of LP-FT, we conduct theoretical analysis by decompling feature extractor and linear head in neural networks. We examine how LP-FT adapts to client-specific local data and its resulting impact on generalization performance. The developed theoretical frameworksheds light on the conditions when LP-FT outperforms full FT-based PFT [1], offering new insights into its efficacy within FL.

In summary, our analysis justifies the superiority of LP-FT over standard fine-tuning methods under different distribution shift scenarios. In this paper, we make the following key contributions: **Evaluation:** We present comprehensive empirical evaluations, spanning seven diverse datasets, various models, and different fine-tuning methods within the PFL framework on various distribution shift settings. Our evaluation exposes key limitations in existing PFT methods, highlighting their tendency to overfit and exhibit inconsistent performance across distribution shifts, despite careful hyperparameter tuning and regularization. **Insight:** We introduce LP-FT, a simple yet effective PFT strategy that enhances both local and global performance. We reveal, for the first time, that preserving pre-trained global features is linked to improved global performance in PFT settings, and we attribute FT's degraded performance to catastrophic feature distortion. **Theory:** We offer a rigorous theoretical analysis of LP-FT using two-layer linear networks, demonstrating its

---

[1]In the following content, we use LP-FT and FT in reference to PFT with LP-FT and full FT, respectively.

superior ability to preserve global performance compared to FT in both concept shift and combined concept-covariate shift scenarios.

## 2 RELATED WORK

**Heterogeneous FL.** Heterogeneous FL refers to a decentralized training paradigm that accommodates diverse and disparate data sources or devices participating in a collaborative model-building process. Examples include FedAvg (McMahan et al., 2017) and various improvements in terms of aggregation optimization and local optimization. It is shown to experience challenges in heterogeneous scenarios. Thus, various literature proposes alternative strategies. Here, we summarize these strategies into aggregation optimization and local optimization. FedNova (Wang et al., 2020) belongs to the category of Aggregation optimization, which normalizes and scales the local updates. Examples of local optimization include FedProx (Li et al., 2020a) and Scaffold (Karimireddy et al., 2020), where FedProx adds a $L_2$ regularization for each client and Scaffold adds a variance reduction term. However, these methods often exhibit limited personalization capabilities and may not adequately meet the performance requirements of different clients. Consequently, various personalized FL approaches have been proposed, with a primary emphasis on enhancing local client performance to the greatest extent possible. We can group these personalized FL strategies into clustering-based methods (Ghosh et al., 2020), transfer learning (Yu et al., 2020), and interpolating the local and global models (Mansour et al., 2020; Deng et al., 2021). Some FL methods (Guo et al., 2024; Son et al., 2024) highlight that existing federated learning methods often fail under feature shift despite addressing label shift, proposing clustering and regularization strategies respectively to tackle diverse distribution shifts in non-IID data settings.

**Regularized Fine-Tuning** Fine-tuning pre-trained models has become increasingly popular with the rise of foundation models (Bommasani et al., 2021). However, fine-tuning with limited data often lead to overfitting. Several strategies can mitigate this issue, such as using optimizers that promote a flatter loss landscape (Li et al., 2018; Kaddour et al., 2022). Notably, Sharpness-Aware Minimization (SAM) (Foret et al., 2021) and Stochastic Weight Averaging (SWA) (Izmailov et al., 2018) are two popular methods that help achieve this. Additionally, a recent technique called *model soups* (Wortsman et al., 2022b), uses a simple greedy weight averaging approach similar to SWA, shown significant improvements in fine-tuning. An interesting perspective focuses on minimizing the linear mode connectivity barrier between the pre-trained and fine-tuned models, helping maintain consistency in decision-making mechanisms from a loss landscape perspective (Vlaar & Frankle, 2022). *Partial fine-tuning* is another common method to prevent overfitting, which involves selectively fine-tuning specific layers of the model to better adapt to variations in data distribution (Lee et al., 2023). Recent studies have introduced the concept of LP-FT (Kumar et al., 2022), highlighting potential distortions in pre-trained features and their underperformance in scenarios involving previously unseen data. Further research on LP-FT provides a deeper analysis of model adaptation (Trivedi et al., 2023), focusing on feature distortion and simplicity bias, thereby enhancing our understanding of fine-tuning mechanisms and safe model adaptation. A detailed literature review is introduced in App. B.

## 3 OBSERVATION: OVERFITTING IN PERSONALIZED FINE-TUNING

In this section, we describe our experimental settings and evaluation methods to analyze personalized overfitting and the effects of LP-FT. This section is organized as follows: (1) defining the problem overview and the distribution shifts considered within a data heterogeneity context, (2) introducing the experimental setups, including the datasets and PFT strategies under investigation, (3) demonstrating the tendency of personalized overfitting with full FT, even with careful regularization, across distribution shifts, (4) proposing LP-FT and evaluating its performance against state-of-the-art full FT strategies in FL, and (5) examining federated feature distortion.

### 3.1 OVERVIEW AND DEFINITIONS

**Problem Setting.** In a FL setting, each client $i \in [C]$ has a local dataset $(\mathbf{X}_i, \mathbf{Y}_i)$ generated from a potentially distinct distribution, which may differ across clients due to distribution shifts. The goal is to perform personalized fine-tuning for each client by optimizing local model parameters $\theta_L$, initialized from a well-trained global model $\theta_G$. The objective is to minimize the local loss $\mathcal{L}_L(\theta_L)$

for improved local performance while ensuring that the global loss $\mathcal{L}_G(\theta_L)$ remains close to that of a pre-trained global model. This creates a trade-off between personalization (minimizing local loss) and maintaining generalization (minimizing global loss) across clients. The global data distribution $\mathcal{D}_G$ is defined as a mixture of the local distributions $\mathcal{D}_i$, given by $\mathcal{D}_G = \frac{1}{C} \sum_{i \in [C]} \mathcal{D}_i$.

We formally define distributions of interests, concept shift and covariate shift in heterogeneous FL context, following Li et al. (2021b).

**Covariate Shift.** Covariate shift refers to variations in the input feature distribution across clients while keeping the conditional distribution of the output given the input consistent. Formally, for any pair of clients $i$ and $j$ with $i \neq j$, the data-generating process is characterized by:

$$P_i(x) \neq P_j(x), \quad \text{but} \quad P_i(y \mid x) = P_j(y \mid x) \quad \text{for all } i \neq j.$$

This means that while clients $i$ and $j$ may have different input distributions $P_i(x)$ and $P_j(x)$, the conditional distribution $P(y \mid x)$ remains consistent across all clients.

**Concept Shift.** Concept shift, in contrast, occurs when the conditional relationship between input features and outputs varies across clients, while the input feature distribution remains unchanged. The data-generating process for any pair of clients $i$ and $j$ with $i \neq j$ is given by:

$$P_i(y \mid x) \neq P_j(y \mid x), \quad \text{but} \quad P_i(x) = P_j(x) \quad \text{for all } i \neq j.$$

In this scenario, all clients share the same input distribution $P(x)$, but the conditional distribution $P_i(y \mid x)$ varies, indicating different relationships between features and labels across clients.

## 3.2 EMPIRICAL ANALYSIS SETTING

**Datasets with Cvariate Shift.** We include `Digit5`, `DomainNet`, `CIFAR10-C`, and `CIFAR100-C`. `Digit5` and `DomainNet` belong to the *feature-shift* subgroup, where the data features represent different subpopulations within the same classes. For example, `Digit5` contains 10-digit images collected from various sources with different backgrounds, such as black-and-white for MNIST and colorful digits for synthetic datasets. `CIFAR10-C` and `CIFAR100-C` belong to the *input-level shift* subgroup, where noise or distortion is introduced in the input data, such as blurred images or sensor errors, degrading input quality. A detailed explanation of the data splitting and its introduction is provided in Table 3 in the Appendix. Visualization is provided in Fig. 5.

**Datasets with Concept Shift.** We include `CheXpert`, and `CelebA`. `CheXpert` and `CelebA` belong to the *spurious correlation-based shift* subgroup, which involves misleading relationships in the training data that models may exploit (also known as spurious correlations), despite being unrelated to the actual target. This reliance can lead to poor performance when such correlations are absent in new data, classifying it as a form of concept shift (Izmailov et al., 2022). Similarly, the detailed information is provided in Table 3 in the Appendix. Visualization is provided in Fig. 5.

**Fine-tuning Strategies.** In this study, we explore several common fine-tuning strategies in PFL: *Full-parameter FT*, a naive FT strategy that adjusts all model parameters. *Proximal FT* (Li et al., 2020b) aims to preserve the pre-trained model's original knowledge by applying proximal regularization, which penalizes large deviations from the initial model parameters to maintain generalization. *Soup FT* (Wortsman et al., 2022a) improves robustness by averaging the weights of multiple fine-tuned model instances, each trained with different initializations, creating a "model soup" that integrates their strengths. Lastly, *Sparse FT* (Lee et al., 2018) promotes sparsity in parameter updates, adjusting only the most relevant weights to enhance efficiency and interpretability, particularly for deployment in resource-constrained environments. Each strategy is designed to balance model performance with different priorities, such as preserving knowledge, enhancing robustness, or improving efficiency. A more detailed experiment setting is presented in App. C.

## 3.3 GLOBAL AND LOCAL PERFORMANCE TRENDS IN COMMON PFT METHODS

In practice, PFT is susceptible to overfitting to local data, due to the relatively small amount of data available at local clients. Unlike conventional overfitting, the overfitting referred to in the FL context is characterized by a consistent improvement in local performance while global performance noticeably deteriorates (Wu et al., 2022; Chen et al., 2023) – the average gain in local performance can be smaller than the loss in global performance. To measure the model's overall local and global

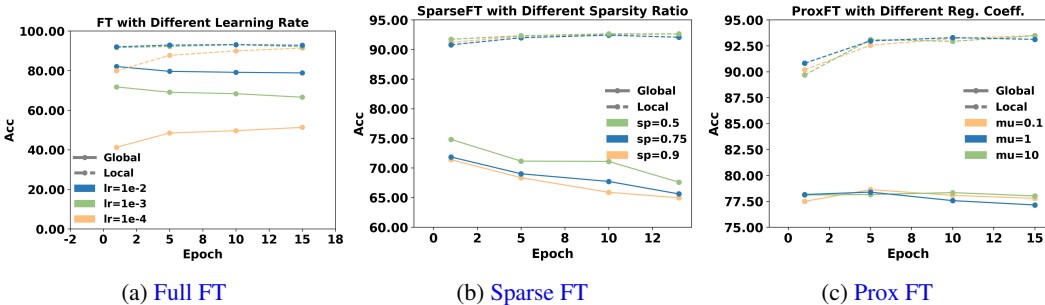

(a) Full FT        (b) Sparse FT        (c) Prox FT

Figure 2: Visualization of the prevalence of personalization overfitting across different distribution shift scenarios, where (a) shows the global and local accuracy under different learning rates for full-parameter fine-tune; (b) shows the different sparsity rate for sparse fine-tune; (c) shows the different regularization strength under the proximal fine-tune. In all subfigures, the global accuracy is shown as the solid line, and the local accuracy is shown as the dashed line. As shown, the global accuracy is kept constant while the local accuracy drops in all hyper-parameter settings. This indicates that the PFT model overgeneralizes due to feature distortion.

performance, we used the averaged client-wise local and global accuracy. The metric's decreasing trend with increasing local training epochs indicates personalized overfitting. Notably, this trend persists even when considering only global performance metrics, as local performance tends to show increases in PFT under overfitting conditions.

In all subplots of Fig. 2, we observe the tendency of common PFT strategies in FL to overfit under various distribution shift conditions, including input-level shifts (`CIFAR100-C`), feature-level shifts (`Digit5`), and spurious correlation-based distribution shifts (`CheXpert`). We systematically adjusted hyperparameters to evaluate their impact on performance. Notably, we tuned the parameters to avoid a significant increase in local performance, ensuring that any decline in the y-axis primarily reflects a drop in global performance. Fig. 2a illustrates that overfitting persists even when fine-tuning methods use SGD optimizers and reduced learning rates. Fig. 2b shows that, in scenarios where the sparsity rate is adjusted (with sparsity involving masking operations on parameter gradients, meaning that higher sparsity rates result in fewer parameters being updated), overfitting still occurs as the number of training epochs increases in sparsity fine-tuning methods. Fig. 2c depicts the average local and global performance of fine-tuning methods using proximal regularization terms. Similarly, even after adjusting the regularization terms to bias updates towards the initial global model, overfitting remains evident. More empirical studies can be found in App. D.

### 3.4 LP-FT: PERFORMANCE COMPARISON ACROSS FINE-TUNING STRATEGIES

**LP-FT.** Observing the challenges of personalized overfitting in common regularized fine-tuning methods in PFT, we propose a straightforward approach called Linear Probing and Subsequent Fine-tuning (LP-FT), which demonstrates strong personalization and generalizability across diverse datasets. *In practice, clients utilize LP-FT by performing linear probing on the global model's linear layers with local data, followed by fine-tuning the full model*. Although the concept of LP-FT was first proposed in centralized setting (Kumar et al., 2022), describing the fine-tuning of a pre-trained model from data in domain $A$ to downstream data in domain $B$, leading to performance degradation on out-of-distribution testing due to domain misalignment, we refer LP-FT differently in PFT. Specifically, the initial pre-trained model is trained on the combined data from all clients (global distribution) and each client has a distinct ground-truth function. This is in contrast to the setting in the ground-truth function, which is the same for all the data points in Kumar et al. (2022). Therefore, it is novel to explore the efficacy of LP-FT in the PFT setting.

**Experimental Settings.** To demonstrate the effectiveness of LP-FT in PFT, we focus on comparing different FT methods at the local PFT stage (see Fig. 1 (a)). We use the FedAvg algorithm to train this shared global model in a GFL manner. After the GFL stage, all the clients further fine-tune the obtained global model using local data for 15 epochs for personalization. The final models are

---
[1]The worst group accuracy on `CIFAR10` is not noted as all baseline FT results are very close to each other, where comparing STD, in this case, is less meaningful.

Table 1: Performance of various PFT strategies. Red represents the *input shift* subgroup; green from the *feature-shift* subgroup; blue the *spurious correlation-based shift* subgroup. Each experiment is performed three times independently with different random seeds, and the standard deviation of the results is presented in parentheses. ↑ indicates that higher values are better, while ↓ indicates that lower values are better.

| Dataset | Method | Local ↑ | Global ↑ | C-Std. ↓ | Worst ↑ | Average ↑ |
|---|---|---|---|---|---|---|
| **CIFAR10-C** | FT | 54.50 (0.64) | 44.16 (0.13) | **10.04 (0.06)** | 19.83 (0.18) | 39.50 (0.33) |
| | Proximal FT | 61.76 (0.13) | 53.58 (0.14) | 11.61 (0.08) | 25.82 (0.12) | 47.05 (0.07) |
| | Soup FT | 56.36 (0.23) | 44.94 (0.06) | 10.22 (0.06) | 20.47 (0.35) | 40.59 (0.09) |
| | Sparse FT | 61.31 (0.01) | 50.21 (0.17) | 11.10 (0.11) | 24.56 (0.09) | 45.36 (0.04) |
| | LP-FT | **63.55 (0.04)** | **55.35 (0.01)** | 12.45 (0.01) | **26.33 (0.06)** | **48.41 (0.03)** |
| **CIFAR100-C** | FT | 20.05 (0.05) | 14.45 (0.04) | **5.37 (0.02)** | 3.37 (0.06) | 12.62 (0.03) |
| | Proximal FT | 27.38 (0.15) | 19.96 (0.11) | 6.90 (0.04) | 4.84 (0.04) | 17.41 (0.05) |
| | Soup FT | 20.99 (0.24) | 14.81 (0.04) | 5.48 (0.03) | 3.56 (0.01) | 13.12 (0.06) |
| | Sparse FT | 28.93 (0.04) | 20.66 (0.02) | 7.75 (0.02) | 5.05 (0.09) | 18.15 (0.10) |
| | LP-FT | **32.60 (0.14)** | **25.44 (0.10)** | 9.66 (0.04) | **5.92 (0.06)** | **21.32 (0.04)** |
| **Digit5** | FT | 91.17 (0.90) | 67.87 (0.74) | 22.93 (0.28) | 42.03 (0.48) | 67.02 (0.70) |
| | Proximal FT | **92.09 (0.18)** | 81.40 (0.03) | 15.04 (0.15) | 61.71 (0.16) | 78.40 (0.09) |
| | Soup FT | 91.82 (0.34) | 70.82 (0.43) | 21.99 (0.67) | 45.10 (1.27) | 69.02 (0.65) |
| | Sparse FT | 91.43 (0.31) | 76.89 (0.72) | 17.90 (0.38) | 54.21 (0.56) | 74.21 (0.35) |
| | LP-FT | 91.20 (0.04) | **82.78 (0.05)** | **13.75 (0.02)** | **65.80 (0.02)** | **79.92 (0.02)** |
| **DomainNet** | FT | 64.90 (1.18) | 42.48 (0.58) | 17.49 (0.75) | 22.31 (0.93) | 43.23 (0.52) |
| | Proximal FT | 67.20 (1.39) | 56.05 (0.27) | **16.68 (0.36)** | 33.20 (1.79) | 52.60 (0.35) |
| | Soup FT | 67.48 (0.61) | 44.27 (0.46) | 18.44 (0.42) | 23.73 (1.24) | 44.49 (0.54) |
| | Sparse FT | **69.62 (0.53)** | 50.24 (0.44) | 18.14 (0.17) | 27.89 (0.15) | 49.14 (0.45) |
| | LP-FT | 68.50 (0.19) | **57.52 (0.20)** | 17.36 (0.21) | **34.53 (0.44)** | **53.52 (0.19)** |
| **CheXpert** | FT | 76.18 (0.41) | 76.25 (0.56) | 0.35 (0.13) | 76.31 (0.76) | 76.25 (0.44) |
| | Proximal FT | 76.44 (0.07) | 76.63 (0.09) | 0.71 (0.09) | 76.81 (0.07) | 76.63 (0.07) |
| | Soup FT | 77.51 (0.15) | 77.49 (0.31) | 0.48 (0.07) | **77.46 (0.43)** | 77.49 (0.26) |
| | Sparse FT | 77.29 (0.13) | 77.20 (0.14) | **0.31 (0.11)** | 77.11 (0.25) | 77.20 (0.14) |
| | LP-FT | **77.64 (0.37)** | **77.54 (0.37)** | 0.53 (0.41) | 77.43 (0.71) | **77.54 (0.37)** |
| **CelebA** | FT | 90.55 (1.20) | 73.76 (2.15) | 18.79 (3.64) | 53.52 (5.51) | 72.39 (2.84) |
| | Proximal FT | **93.74 (0.59)** | 81.11 (0.82) | 13.39 (1.14) | 67.50 (2.10) | 80.78 (0.90) |
| | Soup FT | 89.42 (2.16) | 75.28 (1.11) | 16.29 (1.19) | 57.79 (2.90) | 74.17 (1.50) |
| | Sparse FT | 91.43 (0.48) | 77.32 (1.46) | 14.16 (2.57) | 62.94 (4.34) | 77.65 (1.65) |
| | LP-FT | 93.24 (0.17) | **83.32 (0.31)** | **11.18 (0.14)** | **71.89 (0.75)** | **82.82 (0.64)** |

evaluated using the *metrics* described below. Detailed descriptions of the datasets, preprocessing steps, data splitting, and models used are provided in Appendix C.3, Tab. 3.

**Metrics.** We report five metrics in this part of the experiment: *(1) Local Accuracy (Local)* measures the performance of the PFT model on the client's local test set. Higher *Local Acc* indicates better personalization. *(2) Global Accuracy (Global)* measures the PFT model's average test accuracy over all other clients' test sets. Higher *Global Acc* indicates better generalization. *(3) Client-wise Standard Deviation (C-Std.)* calculates the standard deviation of local test accuracies across all clients. Lower *C-Std.* indicates less variance in performance among clients. *(4) Worst Accuracy (Worst)* reports the lowest test accuracy among all clients. The closer this value is to *Local Acc*, the better the worst-case generalization. *(5) Average* reports the average of both *Local Acc* and *Global Acc*, providing a better understanding of the tradeoff between personalization (local performance) and generalization (global performance). All metrics, except *C-Std.*, are averaged over the number of clients, and higher values are preferable. For the *C-Std.* metric, lower values are better.

**Results.** Our results are presented in Tab. 1, where the best method is highlighted in **bold**. Datasets with the same distribution shift pattern are grouped into the same colors as detailed in the caption. Tab. 1 shows that LP-FT consistently achieves the highest global and average accuracy across most datasets, demonstrating strong generalization and personalization performances, particularly in challenging conditions like `CIFAR100-C` and `CIFAR10-C`. Sparse FT also performs well, especially in `Digits5` and `DomainNet`, but generally lags behind LP-FT. Soup FT and Proximal FT show mixed results, with stronger performance in specific datasets such as `CheXpert` but weaker overall compared to LP-FT. Standard fine-tuning consistently underperforms, highlighting the limitations of basic fine-tuning methods in heterogeneous data scenarios.

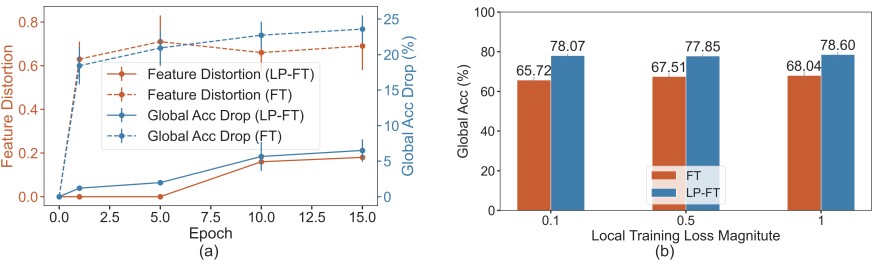

Figure 3: Illustration of federated feature distortion (FD) and decision boundaries.

## 3.5 INSIGHT AND EXPLANATION ON THE OBSERVATIONS

**Federated Feature Distortion.** We hypothesize that personalized overfitting in PFT and the associated performance degradation are linked to federated feature distortion, which refers to the *alteration of intermediate data representations from the global model* (*i.e.*, network features before the linear head) during the final local training phase in PFL, as illustrated in Fig. 1. This distortion, driven by data heterogeneity across clients, occurs when local models overfit to client-specific distributions. To illustrate how feature distortion affects performance, we present Fig. 3, showcasing model decision boundaries and features. Specifically, after GFL, the global server model is well-trained with well-separated decision boundaries for global features. However, it may misclassify some local data points. While fine-tuning local clients with full FT can improve local performance, it severely distorts their feature representations compared to those from global model, potentially resulting in shifted decision boundaries that harm global performance. In contrast, LP-FT uses linear probing first, where only the linear head is tuned, preserving the pre-trained feature representations. The subsequent fine-tuning after linear probing causes only minor adjustments to full-model parameters and features, as the model is already close to optimal for the local data. This process can result in moderate feature distortion, simultaneously fitting the model to local data while maintaining global performance.

**Experimental Validation on the Relationship Between Feature Distortion and Performance.** In this part, we utilize multiple datasets (*i.e.*, `DomainNet` and `Digit5`) to investigate the relationship between global performance and feature distortion. We measure the similarity of representations in the feature extraction layer (*i.e.*, the input to the classification head) in the following way. Consider a feature extraction function $f : \mathcal{X} \to \mathbb{R}^k$, which maps inputs from the input space $\mathcal{X}$ to a representation space $\mathbb{R}^k$. Let $\theta_G$ denote the global pre-trained model and $\theta_i$ the fine-tuned model after local fine-tuning for client $i$. Assume there are $C$ clients in total, each with $n$ samples. Let $x_{c,j}$ represent the $j$-th data point of the $c$-th client. The *federated feature distortion* $\Delta_c(f)$ quantifies the change in features after fine-tuning for the $c$-th client, defined as the average $\ell_2$ distance between the representations produced by the global model and the locally fine-tuned model over all data points across all clients. Formally, it is expressed as: $\Delta_c(f) = \frac{1}{n} \sum_{j=1}^{n} \| f(\theta_G; x_{c,j}) - f(\theta_c; x_{c,j}) \|_2$, where $\| \cdot \|_2$ is the $\ell_2$ distance in the representation space $\mathbb{R}^k$. We compute the average of $\Delta_c(f)$ across all clients to represent the feature distortion in the PFT setting, as shown in Fig. 4.

*Observation 1.* Our findings reveal that feature distortion is correlated with a drop in global performance. Specifically, in Fig. 4(a), common FT methods cause significant feature distortion, leading

Figure 4: Observations of the feature distortion in our PFT setting, where (a) presents the positive correlation between global performance drops and feature distortion intensity on `DomainNet` and (b) presents the ablation study on preserving federated features with controlled local train loss on `Digit5`. We set local loss thresholds (0.1, 0.5, and 1.0) and used gradient ascent when the loss fell below, ensuring training loss fluctuated around these points.

to a substantial decline in global performance. In contrast, LP-FT maintains a high level of global performance. This stability can be attributed to the initial phase of LP-FT (the first 5 epochs), during which only the classifier is fine-tuned while the parameters of the feature extraction layer remain fixed. Compared with FT, LP-FT suffers from significantly moderate feature distortion.

*Observation 2.* In Fig. 4(b), we further analyze the relationship between global performance and feature distortion by eliminating the influence of local loss magnitude. We achieve this by controlling the level of local training loss using the loss flooding technique (Ishida et al., 2020). Specifically, we set thresholds for local loss (*i.e.*, 0.1, 0.5, and 1.0) and apply gradient ascent when the loss drops below these thresholds, ensuring that the training loss fluctuates around the set points. Examining a single FT strategy (*i.e.*, either FT or LP-FT), we observe that global accuracy remains stable despite variations in local training loss magnitudes. At fixed local training loss levels, we compare the effects of LP-FT (moderate feature distortion) and FT (severe feature distortion) on global performance. LP-FT consistently delivers better global performance than FT across different loss levels. These comparisons enable us to dismiss the alternative explanation that FT leads to higher global loss simply due to achieving lower local loss, while LP-FT exhibits the opposite pattern.

# 4 THEORETICAL ANALYSIS OF THE LP-FT METHOD

In Sec. 3, we presented a series of experiments demonstrating the effectiveness of LP-FT compared to common FT strategies in PFL. Our results indicate that FT performs poorly relative to LP-FT, and we hypothesize that this suboptimal global performance arises from federated feature distortion. To further understand how feature learning impacts generalization error in PFT, we decompose the data-generating function and the model into two components: a feature extractor and a linear head. This decomposition allows us to distinguish between the learned features and their influence on performance. To further explain the empirical observations in Sec. 3, we provide a theoretical analysis of LP-FT's global performance under various distribution shift scenarios. Specifically, in Sec. 4.1 and Sec. 4.2, we formalize concept and covariate shifts within a two-layer linear network and examine how LP-FT effectively adapts to these shifts, outperforming FT in the PFL setting.

**Overview of Theoretical Analysis:** To compare the performance of LP-FT and FT, we make assumptions about the data-generating function for clients (Assumption 4.1) and a specific model structure (Assumption 4.2). Based on these assumptions, we analyze the global performance of LP-FT and FT under concept shift (Theorem 4.4) and combined concept-covariate shift (Theorem 4.5).

## 4.1 LP-FT'S GLOBAL PERFORMANCE UNDER CONCEPT SHIFT

In this section, we analyze LP-FT's performance compared to FT under concept shift. To facilitate a rigorous theoretical study, we define the data-generating process and model structure across clients, assuming both are represented by two-layer linear networks, as in (Kumar et al., 2022).

**Assumption 4.1** (Data-Generating Process)**.** The data-generating function for client $i$ is given by $y_i = V_i^{*T} B_* x_i$ for all $i \in [C]$, where $y_i \in \mathbb{R}$, $C$ is the number of clients, $x_i \in \mathbb{R}^d$, $B_* \in \mathbb{R}^{k \times d}$, and $V_i^* \in \mathbb{R}^k$. All clients share a common feature extractor $B_*$, assumed to have orthonormal rows, while their linear heads $V_i^*$ differ. Each $V_i^*$ decomposes as $V_i^* = \begin{bmatrix} V_{com}^{*T} & \lambda e_i^T \end{bmatrix}^T$, where $V_{com}^* \in \mathbb{R}^m$ is shared across clients, $e_i \in \mathbb{R}^C$ is a unit vector, and $\lambda$ controls heterogeneity. Here, $m + C = k$.

This assumption distinguishes between a shared and client-specific component in the data-generating functions, allowing analysis of both global and local performance of PFT methods after fine-tuning.

**Assumption 4.2** (Model Structure)**.** The training model is a two-layer linear network defined as $y = V^T B x$, where $V \in \mathbb{R}^k$ is the linear head and $B \in \mathbb{R}^{k \times d}$ is the feature extractor. The dimensions of $V$ and $B$ match Assumption 4.1, allowing the model to learn both shared and client-specific data components.

In PFT settings, our objective is to evaluate the performance of a model on both global and local data. By local data, we refer to the data of a specific client undergoing fine-tuning (e.g., client $i$). The local

and global losses are defined using the Mean Squared Error (MSE) as follows:

$$\mathcal{L}_L(V, B) = \mathbb{E}_{(x,y)\sim\mathcal{D}_i}\left[\frac{1}{2}(V^T Bx - y)^2\right] = \mathbb{E}_{x\sim\mathcal{D}_i}\left[\frac{1}{2}(V^T Bx - V_*^{*T} B_* x)^2\right],$$

$$\mathcal{L}_G(V, B) = \mathbb{E}_{(x,y)\sim\mathcal{D}_G}\left[\frac{1}{2}(V^T Bx - y)^2\right] = \frac{1}{C}\sum_{i\in[C]}\mathbb{E}_{x\sim\mathcal{D}_i}\left[\frac{1}{2}(V^T Bx - V_i^{*T} B_* x)^2\right].$$

Since this section focuses on concept shift, we assume all clients' data is drawn from similar distributions. Accordingly, we assume for every client $i \in [C]$, the input features satisfy $\mathbb{E}_{x\sim\mathcal{D}_i}[xx^T] = I_d$.

With the theoretical framework established by Assumptions 4.1 and 4.2, we compare the global performance of LP-FT and FT, highlighting cases where LP-FT outperforms FT. In a PFL setting, the initial model is trained on data from all clients to capture their shared components. Thus, we initialize the model parameters as $B_0 = B_*$ and $V_0 = \begin{bmatrix} V_{com}^{*}{}^T & \mathbf{0} \end{bmatrix}^T$. In LP-FT, a step of linear probing first updates $V_0$ using local data while keeping $B_0$ fixed, followed by full fine-tuning to update both $V$ and $B$. In contrast, FT performs only the second step. The following lemma characterizes $B$ after one gradient descent step in FT, forming the basis for our comparison.

**Lemma 4.3.** *Under Assumptions 4.1 and 4.2, and assuming that $\mathbb{E}_{x\sim\mathcal{D}_i}[xx^T] = I_d$ for all clients $i \in [C]$, let the initial parameters before starting FT be $B_0 = B_*$ and $V_0 = \begin{bmatrix} V_{com}^{*}{}^T & \mathbf{0} \end{bmatrix}^T$. Assume fine-tuning is performed locally on the data of the $i$-th client. Let $B_{FT}$ denote the feature extractor matrix after a single gradient descent step (processing the entire dataset once) with learning rate $\eta$. If $(b_j^{FT})^T$ is the $j$-th row of $B_{FT}$, then:*

$$\mathbb{E}\left[(b_j^{FT})^T\right] = (b_j^*)^T + \eta\lambda(V_0)_j(b_{m+i}^*)^T,$$

*where $(b_j^*)^T$ is the $j$-th row of $B_*$, and $(V_0)_j$ is the $j$-th element of $V_0$ for $j \in [k]$.*

This lemma examines the impact of FT on the feature extractor $B_{FT}$, highlighting the deviations from the pre-trained matrix $B_0 = B_*$. Given that all clients share the same $B_*$ in their labeling functions, substantial changes to the feature extractor can lead to a decline in global performance. Since the matrix $B$ functions as the feature extractor in our framework, significant feature distortion occurs when $B_{FT}$ deviates considerably from $B_*$. Building on Lemma 4.3, Theorem 4.4 offers a comparative analysis of the global performance of LP-FT versus FT in the context of concept shift.

**Theorem 4.4.** *Under Assumptions 4.1 and 4.2, and assuming $\mathbb{E}_{x\sim\mathcal{D}_i}[xx^T] = I_d$ for all clients $i \in [C]$, let the initial model parameters be $B_0 = B_*$ and $V_0 = \begin{bmatrix} V_{com}^{*}{}^T & \mathbf{0} \end{bmatrix}^T$. Let $B_{FT}$ and $V_{FT}$ denote the parameters of the FT method after one gradient descent step (processing the entire dataset once). For LP-FT, let $B_{LPFT}$ and $V_{LPFT}$ denote the parameters after (i) linear probing, which optimizes $V$ with $B$ fixed at $B_*$, and (ii) one gradient descent step with learning rate $\eta$. Then:*

$$\mathcal{L}_G(V_{LPFT}, B_{LPFT}) \leq \mathcal{L}_G(V_{FT}, B_{FT}).$$

This theorem characterizes the global performance of LP-FT, suggesting that under concept shift, LP-FT achieves better performance on global data than FT. When starting from a model initialized to capture the shared feature extractor and linear head among clients, LP-FT is more effective in minimizing global loss, aligning with common FL scenarios where the initial model leverages shared client structure.

### 4.2 LP-FT's Global Performance under Combined Concept and Covariate Shifts

In the previous section, we assumed all clients' data came from the same distribution with $\mathbb{E}_{x\sim\mathcal{D}_i}[xx^T] = I_d$. However, this may not hold in many practical scenarios. To address this, we introduce covariate shift, where each client's data is generated as $x_i = e_i + \epsilon n$, with $n \sim \mathcal{N}(0, I)$, $e_i$ as a client-specific shift, and $\epsilon$ controlling the noise level. The model structure and data-generating assumptions remain consistent with Sec. 4.1. This section thus considers both concept and covariate shifts. Theorem 4.5 analyzes the impact of heterogeneity on the global performance of LP-FT and FT.

**Theorem 4.5.** *Under Assumptions 4.1 and 4.2, let each client's data be $x_i = e_i + \epsilon n$, where $n \sim \mathcal{N}(0, I)$ and $e_i$ is a client-specific shift. Assume the initial parameters are $B_0 = B_*$ and $V_0 = \begin{bmatrix} V_{com}^{*}{}^T & \mathbf{0} \end{bmatrix}^T$. Let $B_{FT}, V_{FT}$ be the FT parameters after one gradient descent step, and*

$B_{LPFT}, V_{LPFT}$ *be the LP-FT parameters after linear probing and one gradient descent step (with learning rate $\eta$). Then, there exists a threshold $\lambda^*$ such that for all $\lambda \leq \lambda^*$:*

$$\mathcal{L}_G(V_{LPFT}, B_{LPFT}) \leq \mathcal{L}_G(V_{FT}, B_{FT}).$$

*Remark* 4.6. In Theorem 4.5, the parameter $\lambda$ characterizes the level of heterogeneity among clients. The theorem shows that under both covariate and concept shifts, LP-FT outperforms FT in low heterogeneity settings ($\lambda \leq \lambda^*$), highlighting its advantage in maintaining generalization. To further reinforce the theoretical insights and cover more extensive settings, Sec. 5 provides empirical validation of our findings, confirming the global superiority of LP-FT over FT. While our theoretical analysis of Theorem 4.5 focuses on the low heterogeneity regime, the experiments in Sec. 5 explore a broader range, simulating both high and low heterogeneity levels. These results validate and extend our theoretical insights, demonstrating that LP-FT consistently outperforms FT across all heterogeneity levels (see also Sec. G in the appendix).

## 5 EXPERIMENT: FURTHER VALIDATIONS FOR THEORETICAL FINDINGS

Despite being based on simplified data and model assumptions, our theoretical results demonstrate significant practical relevance. In this section, we empirically validate the contributions in Sec. 4, exploring the performance implications of controllable heterogeneities in neural networks and datasets.

**Experimental Settings.** To validate the impact of $\lambda$ in Theorem 4.5, we simulate a controllable concept shift setting on the `Digit5` dataset with label-flipping under PFT for both FT and LP-FT. For each client, a proportion of labels is randomly flipped, referred to as the flipping ratio. For example, class one is flipped with class two for the first client, and class two with class three for the second, using a randomized mechanism. A higher flipping ratio indicates greater heterogeneity $\lambda$. The settings align with prior studies: the model is pre-trained within the FL framework and used to initialize both FT and LP-FT. This simulates the combined concept-covariate shift discussed in Sec. 4.2. Flipping labels reflects different labeling functions, where higher flipping rates indicate stronger concept shifts. The `Digit5` dataset also introduces covariate shift, as outlined in Sec. 3.2.

Table 2: Performance under Label-Flipping for FT and LPFT, with LF.R. as the label-flipping ratio.

| LF.R. (%) | Metric | FT | LPFT |
|---|---|---|---|
| 20 | **Avg.** ↑ | 67.73 | 79.83 |
|  | **Global** ↑ | 68.76 | 83.08 |
|  | **Local** ↑ | 91.32 | 91.23 |
| 30 | **Avg.** ↑ | 60.04 | 72.95 |
|  | **Global** ↑ | 55.18 | 72.75 |
|  | **Local** ↑ | 91.12 | 89.20 |
| 40 | **Avg.** ↑ | 58.27 | 71.55 |
|  | **Global** ↑ | 53.70 | 69.89 |
|  | **Local** ↑ | 90.84 | 90.02 |
| 50 | **Avg.** ↑ | 60.06 | 73.26 |
|  | **Global** ↑ | 56.17 | 72.32 |
|  | **Local** ↑ | 91.88 | 90.87 |

**Results.** As shown in Tab. 2, LP-FT consistently outperforms FT in global performance across various flipping ratios. This aligns with our theoretical results in Sec. 4.2, especially for deep neural networks under realistic PFT settings. The flipping rate controls concept shift heterogeneity, with higher rates indicating greater heterogeneity, while varying data distributions introduce covariate shift. These experiments simulate the combined concept-covariate shift, as analyzed in our framework. Notably, LP-FT outperforms FT in all heterogeneity levels, validating its advantage in both low and high heterogeneity regimes (larger flipping ratios).

## 6 CONCLUSION

In this work, we tackled the key challenge of balancing local personalization and global generalization in PFL. Through an extensive empirical evaluation across seven datasets, multiple model architectures, and various distribution shifts, we revealed critical limitations in existing PFT methods, which often suffer from overfitting and inconsistent performance across scenarios. To address these issues, we introduced a simple yet effective strategy combining Linear Probing with full Fine-Tuning (LP-FT), which consistently outperforms other methods by preserving pre-trained global features and mitigating the adverse effects of excessive personalization. Through in-depth analysis, we attribute LP-FT's strong performance to its ability to prevent feature distortion, which we linked to the degradation of global performance. Furthermore, we provided a theoretical analysis using two-layer linear networks, explaining LP-FT's superior performance, particularly under concept and covariate shift conditions.

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

## A    Navigating the Trade-Off Between Local and Global Performance in FL

**The Challenge of improving both local and global performance.**    In FL, a primary challenge in enhancing both local and global performance arises from data heterogeneity. The disparities in data distribution among individual clients result in divergent risk functions for each client, consequently leading to disparate optimal solutions. Additionally, in the context of federated learning, each client's local training process is oblivious to the data of other clients, rendering models prone to overfitting to local data distribution after personalized fine-tuning. This personalization overfitting phenomenon detrimentally impacts global performance.

Previous approaches to personalized federated learning, primarily based on local training, can be broadly categorized into two main types: Partial fine-tuning (which fine-tunes only specific layers of the global model on local data to retain globally learned features) (Collins et al., 2021; Arivazhagan et al., 2019), and regularization guided by the global model to constrain local updates (Li et al., 2021a; Deng et al., 2020). However, while mitigating overfitting to some extent during local training, these methods often face challenges in significantly improving both local and global performance, especially in scenarios with small local training datasets and increased data heterogeneity, as observed in (Wu et al., 2022).

**Potential of improving the trade-off of local and global performance from the perspective of loss landscape.**    In order to better understand the dilemma of simultaneously improving local and global performance, we provide a novel perspective based on the analysis of personalized overfitting using loss landscape. We believe that the fundamental reason for overfitting in previous PFL methods is the inability to find a wide low-loss basin influenced by both local and global losses. Previous GFL methods focus solely on finding the optimal solution in the global loss landscape, while PFL methods only focus on the optimal solution in the local loss landscape, neglecting the structural information of the combined loss landscape composed of the two losses. Due to lacking consideration of the offset between the global and local loss landscapes caused by data heterogeneity, even if previous PFL methods can find a wide low-loss basin in the local loss landscape, they still cannot guarantee generalization to other clients. That is also the reason why personalization overfitting emerges.

## B    Fine-tune Details

This section provides an overview of the baseline techniques utilized in our study. We describe the characteristics and implementation specifics of three main fine-tuning methods: Proximal FT, Soup FT, and Sparse FT.

### B.1    Proximal FT

Proximal Fine-Tuning (Proximal FT) (Li et al., 2020b) is a method that emphasizes preserving the original knowledge of the pre-trained model while adapting it to new tasks. This technique employs proximal regularization, which penalizes large deviations from the initial model parameters during the fine-tuning process. The primary advantage of Proximal FT is its ability to maintain the generalization capabilities of the pre-trained model, thus reducing the risk of overfitting to the new task's data. In our experiments, we used an L2 regularization term to enforce proximity between the pre-trained and fine-tuned weights, with a regularization coefficient of 0.01.

### B.2    Soup FT

Soup Fine-Tuning (Soup FT) (Wortsman et al., 2022a) is an innovative approach that leverages the concept of "model soups," where multiple fine-tuned models are combined to create a more robust final model. The key idea is to fine-tune several instances of the pre-trained model on the target task with different random initializations or data shuffling, and then average the resulting weights to form a "soup." This method aims to enhance model robustness and performance by integrating the strengths of various fine-tuning instances. For our implementation, we fine-tuned five versions of the pre-trained model and averaged their parameters to create the final Soup FT model.

### B.3   SPARSE FT

Sparse Fine-Tuning (Sparse FT) (Lee et al., 2018) introduces sparsity constraints into the fine-tuning process, encouraging the model to update only a subset of its parameters. This approach aims to improve model efficiency and interpretability by ensuring that only the most relevant weights are adjusted during training. Sparse FT can be particularly beneficial for deploying models in resource-constrained environments where computational efficiency is paramount. In our experiments, we applied L1 regularization to enforce sparsity, setting the regularization coefficient to 0.001 to achieve a balance between performance and sparsity.

### B.4   LP-FT

Linear Probing and then Fine-Tuning (LP-FT) (Kumar et al., 2022) is a two-step transfer learning approach designed to balance in-distribution (ID) and out-of-distribution (OOD) performance. In the first step, linear probing trains only the final layer (head) while freezing the pretrained feature extractor to ensure OOD robustness. The second step fine-tunes all model parameters to improve ID accuracy while retaining the benefits of linear probing for OOD generalization. LP-FT addresses the trade-offs inherent in full fine-tuning by initializing with a well-aligned linear head, reducing feature distortion during optimization. Empirically, LP-FT demonstrates superior ID and OOD accuracy across diverse datasets.

## C   EXPERIMENTAL DETAILS

### C.1   COMPUTING ENVIRONMENT AND HYPER-PARAMETERS

All experiments in this paper are conducted on NVIDIA A40 Graphics cards using PyTorch. The Adam optimizer is employed with a learning rate of $1 \times 10^{-3}$. In FL for all datasets, the standard local model update epochs are set to 1. The communication round is set to be 100 epochs, where we validated the model results from FL converged. Unless specified otherwise, the batch size for all benchmarks is standardized at 128. To ensure a fair comparison with various baselines, all methods initiate the FL personalized fine-tuning with models derived from the best-performing global model in terms of overall effectiveness.

### C.2   VISUALIZATION OF THE ORIGINAL IMAGES

Fig. 5 illustrates a visual representation of the various datasets used in this study, categorized by their levels of transformation or domain. The figure is divided into three main sections:

(a) *Feature-Level Shift (Digit5 and DomainNet):* The left panel displays examples from the Digit5 dataset, showcasing digit images in diverse styles and appearances. These include handwritten digits, digits rendered in varying fonts, and those with unique textures. The right panel features images from the DomainNet dataset, which includes objects and scenes represented in various artistic styles such as clip art, sketches, and realistic photographs. Examples include a strawberry, a zebra, and a cat.

(b) *Input-Level Shift (CIFAR10-C and CIFAR100-C):* This section highlights images from CIFAR10-C and CIFAR100-C, which apply corruptions to standard CIFAR datasets to evaluate robustness. Corruptions include noise, blurring, and distortions that affect the clarity and quality of the images. Example categories feature animals, vehicles, and natural scenes under different types of degradation.

(c) *Output-Level Shift (CheXpert and CelebA):* The left panel presents grayscale X-ray images from the CheXpert dataset, widely used in medical imaging tasks. The right panel showcases color images of celebrity faces from the CelebA dataset, commonly used for facial recognition and attribute prediction.

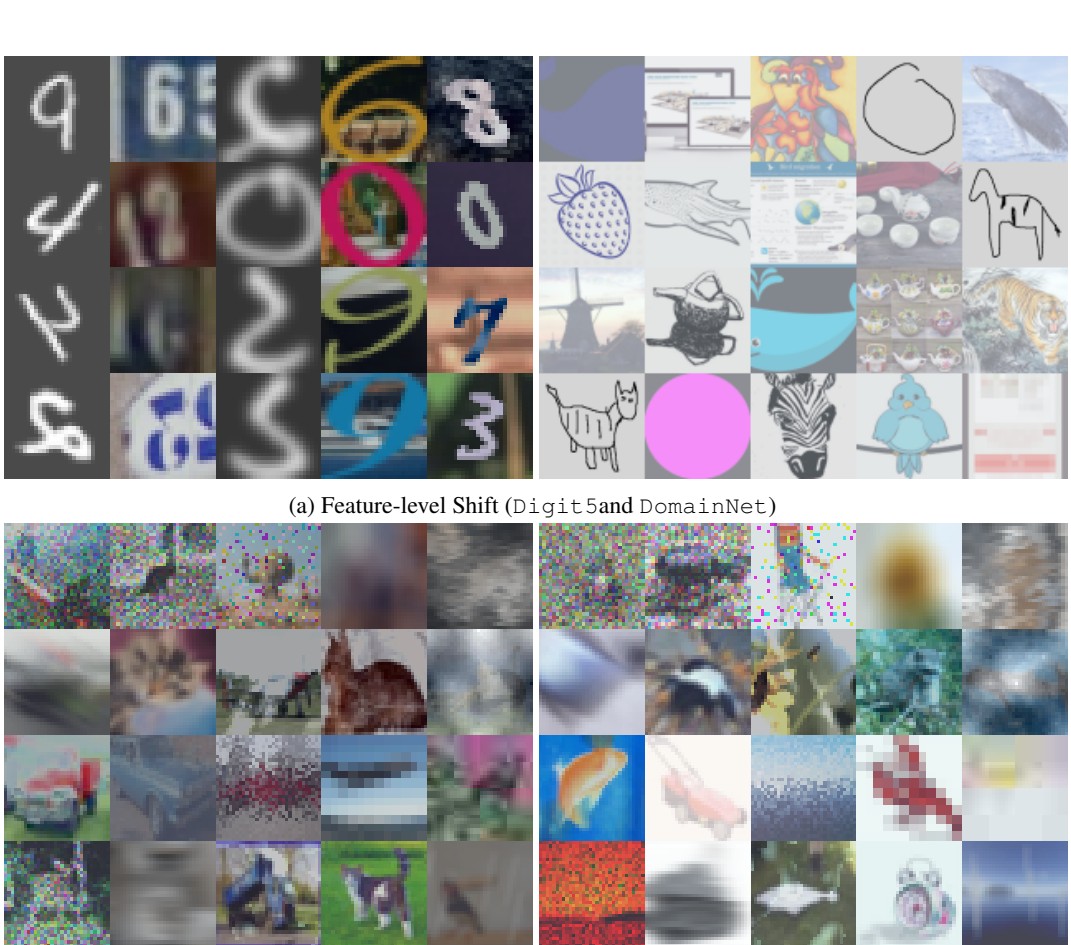

(a) Feature-level Shift (`Digit5`and `DomainNet`)

(b) Input-level Shift (`CIFAR10-C` and `CIFAR100-C`)

(c) Output-level Shift (`ChexPert` and `CelebA`)

Figure 5: Visualization of the original datasets used in the paper.

## C.3 DETAILED DATASET AND MODEL INFORMATION

**Tab. 3** provides a visual overview of the datasets used in this study, categorized by their levels of transformation and data heterogeneity. The table is divided into four sections, corresponding to feature-level, input-level, output-level, and label shift settings:

*Feature-Level Shift (Digit5 and DomainNet):* The **Digit5** dataset, which consists of digit images collected from five distinct domains, including MNIST, SVHN, USPS, SynthDigits, and MNIST-M.

These images exhibit a variety of styles, such as handwritten digits, digits rendered in different fonts, and textured representations, demonstrating substantial visual heterogeneity. The **DomainNet** dataset, a large-scale collection featuring objects and scenes from six domains, including styles like clip art, sketches, and realistic photographs.

*Input-Level Shift (CIFAR10-C and CIFAR100-C):* This type of distribution shift includes corrupted versions of CIFAR-10 and CIFAR-100 datasets. The **CIFAR10-C** dataset applies 50 types of corruptions, such as noise, blur, and distortions, to evaluate model robustness under various degradation conditions. Similarly, **CIFAR100-C** extends the CIFAR-100 dataset by introducing the same set of corruptions, enabling robustness evaluation on a larger and more diverse set of categories.

| Data Heterogeneity Type | Dataset | Model | Description | Clients | Classes |
|---|---|---|---|---|---|
| **Feature-Level Shift** | Digit5 | ResNet18 | A collection of digit images from five domains, used for domain adaptation and digit recognition tasks (Huang et al., 2023). Datasets include MNIST, SVHN, USPS, SynthDigits, and MNIST-M. | 5 | 10 |
| | DomainNet | ResNet50 | A large-scale dataset of images from six distinct domains for multi-source domain adaptation (Peng et al., 2019b). Preprocessing follows the strategy in FedBN (Li et al., 2021c). | 6 | 10 |
| **Input-Level Shift** | CIFAR10-C | ResNet18 | A corruption benchmark dataset for CIFAR-10 (Hendrycks & Dietterich, 2019), augmented with 30 additional corruption types (Mintun et al., 2021) and one extra type (Chen et al., 2021), totaling 50 corruption types (including the original 19 types of corruption from CIFAR-10-C, plus 30 additional corruption types and one extra type, resulting in a total of 50 distinct corruption types). | 50 | 10 |
| | CIFAR100-C | ResNet18 | An extension of CIFAR-100 with common corruptions, following the same strategy as CIFAR-10-C. | 50 | 100 |
| **Output-Level Shift** | CheXpert | ResNet50 | A chest radiograph dataset labeled for 14 common chest conditions (Irvin et al., 2019). Edema and No Finding labels are grouped as described in (Jin et al., 2024). Clients are spuriously correlated with the attribute gender (e.g., 90% of label 1 examples in a client are male). | 2 | 2 |
| | CelebA | ResNet50 | Over 200,000 celebrity images with 40 attributes (Liu et al., 2015). Client splitting follows the same strategy as CheXpert, with attributes: male, female, blonde hair, and non-blonde hair. | 4 | 2 |
| **Label Shift** | CIFAR10 | ResNet18 | A benchmark dataset with label shift induced via Dirichlet distribution ($\alpha = 0.1$), distributed across 20 clients (Krizhevsky et al., 2009). | 20 | 10 |

Table 3: Detailed information about the datasets and their splitting strategies used in the study. For all the settings above, each client has an individual data distribution, ensuring the non-IID nature required for heterogeneous federated learning. Feature-level shift, also referred to as subgroup shift, and input-level shift, corresponding to image corruption, are categorized as covariate shift. Output-level shift, representing spurious correlations in our setting, is categorized as concept shift.

*Output-Level Shift (CheXpert and CelebA):* The **CheXpert** dataset, a widely used medical imaging dataset labeled for 14 common chest conditions. In this study, the Edema and No Finding labels are grouped, and spurious correlations are introduced at the client level, where attributes (i.e. gender in our client splitting) are disproportionately represented (i.e., 90% of label 1 examples in a client are a certain attribute). The **CelebA** dataset, which includes over 200,000 celebrity faces annotated with 40 attributes. Client splitting follows the same strategy as CheXpert, with attributes such as male, female, blonde hair, and non-blonde hair used to create spurious correlations across clients.

*Label Shift (CIFAR10):* The final one focuses on label shift in the **CIFAR10** dataset. This setting simulates non-IID conditions by inducing label distributions across clients using a Dirichlet distribution with $\alpha = 0.1$. This creates significant variations in class distributions among the 20 clients, mimicking real-world federated learning scenarios where data availability across clients is inherently imbalanced.

# D    FURTHER EMPIRICAL RESULTS

## D.1    OTHER PEFT METHODS DISTORT FEDERATED FEATURE

| PEFT Method | Local | Global | Avg. |
|---|---|---|---|
| **LoRA (lr=1e-3)** | 41.54 | 26.75 | 26.87 |
| **LoRA (lr=1e-4)** | 42.01 | 26.41 | 27.18 |
| **Adapter (lr=1e-3)** | 48.35 | 39.28 | 38.08 |
| **Adapter (lr=1e-4)** | 49.07 | 39.83 | 38.36 |
| **LP-FT** | 68.50 | 57.52 | 53.52 |

Table 4: Different PEFT compared on DomainNet with ViT.

**Setup.**   In this study, we adapted two widely recognized parameter-efficient fine-tuning (PEFT) methods: LoRA (Hu et al., 2022) and Adapter (Houlsby et al., 2019). Both methods were fine-tuned with meticulous adjustments to their learning rates on the ViT, as detailed in Tab. 4. These configurations allowed us to assess the impact of different fine-tuning strategies on federated features. The performance of each method was measured in terms of local, global, and average accuracy.

**Result.**   The effectiveness of bias tuning in personalized fine-tuning naturally raises the question of whether other PEFT methods, commonly used for fine-tuning large models, exhibit similar effectiveness in our setting. In this study, we compare the local and global performance of other popular PEFT methods. Our findings reveal that while these methods can achieve high levels of local performance, their global performance still drops significantly, indicating that they distort the federated features to a certain extent. These PEFT methods' local and global performance still fall short compared to LP-FT, indicating that they distort the federated features to a certain extent.

The comparison of PEFT methods in the context of personalized fine-tuning sheds light on the unique challenges and requirements of this setting. Despite the success of PEFT techniques in fine-tuning large models for various tasks, our results suggest that their direct application to personalized FL may not yield optimal results in terms of preserving the global knowledge captured by the federated features.

## D.2    PFL RESULTS

**Setup.**   To further demonstrate the importance of preserving federated features, we compare LP-FT with other popular personalized FL methods. Our primary focus in this paper is on the training during the FT phase. Unless otherwise specified, we consistently use models trained with FedAvg for FL. Unlike other personalized FL methods, which often involve additional operations during the FL phase (such as local training or model aggregation), our LP-FT method relies solely on vanilla FedAvg.

**Results.**   From the table, it is evident that even without additional operations, LP-FT, which focuses on preserving federated features, remains highly competitive compared to other personalized FL methods. This observation underscores the significance of retaining the global features learned during the FL process. By directly comparing LP-FT with methods that employ extra techniques during the federated training phase, we demonstrate that the simple yet effective approach of preserving federated features can yield competitive results. This finding suggests that the key to achieving strong personalization lies in maintaining the knowledge acquired through collaboration across multiple clients, rather than relying on complex modifications to the FL algorithm.

| PFL Method | Local Acc. (↑) | Global Acc. (↑) | Avg. Acc. |
|---|---|---|---|
| FedBN (Li et al., 2021b) | 88.68 | 61.35 | 61.00 |
| PerAvg (Fallah et al., 2020) | 87.06 | 67.26 | 66.66 |
| FedNova (Wang et al., 2020) | 88.68 | 54.26 | 53.41 |
| FedRep (Collins et al., 2021) | 86.94 | 52.99 | 52.57 |
| FedSoup (Chen et al., 2023) | 90.30 | 75.62 | 75.21 |
| LP-FT | 93.03 | **82.46** | **82.17** |

Table 5: Different PFL Methods on CelebA.

## D.3 LABEL SHIFT

We simulated label shift using a Dirichlet distribution with an alpha parameter set to 0.1. The dataset was distributed across 20 clients, and we trained a ResNet18 model for classification. Results is shown in Tab. 6.

| | Baseline | Local | Global | C-Std. | Worst | Avg |
|---|---|---|---|---|---|---|
| | FT | 87.20 (0.24) | 16.67 (0.14) | **26.81 (1.47)** | 0.01 (0.00) | 34.62 (0.09) |
| **CIFAR10** | Proximal FT | 89.80 (1.21) | 17.42 (1.46) | 27.31 (0.08) | 0.01 (0.00) | 35.75 (0.09) |
| | Soup FT | 88.16 (0.15) | 16.94 (0.05) | 27.07 (0.11) | 0.01 (0.00) | 35.04 (0.04) |
| | Sparse FT | 89.16 (0.00) | 17.54 (0.14) | 27.27 (0.02) | 0.01 (0.00) | 35.57 (0.04) |
| | LP-FT | **90.15 (0.43)** | **17.73 (0.16)** | 27.37 (0.09) | 0.01 (0.00) | **35.96 (0.10)** |

Table 6: Comparison of PFT methods on CIFAR10 label shift setting.

Tab. 6 compares different fine-tuning methods on CIFAR10 across multiple evaluation metrics, including local performance, global performance, robustness to corruption (C-Std.), worst-case performance, and average performance. Among the methods, LP-FT achieves the best results, excelling in local performance (90.15), global evaluation (17.73), and average performance (35.96). Proximal FT and Sparse FT also perform competitively, with improvements over standard fine-tuning (FT) and Soup FT in most metrics. All methods show near-identical performance in the worst-case scenario (0.01), indicating a shared limitation in extreme cases. Overall, LP-FT demonstrates the most robust and effective fine-tuning approach on CIFAR10.

## E LIMITATIONS AND BROADER IMPACT.

In our paper, we focused exclusively on vision-related tasks. Extending our empirical findings to language tasks or multimodal scenarios would be a promising direction for future research. Our FL study primarily addresses simulated cross-silo FL settings. Validating our conclusions in real-world FL deployments would also be a worthwhile direction for future exploration. Deploying FL that balances personalization and generalization capabilities in healthcare scenarios holds great promise. However, excessive personalization could lead to issues with the fairness of algorithmic decisions.

## F PROOFS

*Proof of Lemma 4.3.* We want to analyze the Fine-Tuning (FT) method, focusing on the effect of initial parameters. We perform one pass through the entire dataset to simulate the complete fine-tuning process. Consider the Mean Squared Error (MSE) loss function with parameters $V$ and $B$, where $B$ is represented as follows:

$$
B = \begin{bmatrix} b_1^T \\ \vdots \\ b_m^T \\ b_{m+1}^T \\ \vdots \\ b_{m+C}^T \end{bmatrix},
$$

where $b_i^T \in \mathbb{R}^{1 \times d}$ denotes the $i$-th row of matrix $B$, and $m + C = k$.

To apply one step of gradient descent, we need to compute the gradient of the loss function with respect to $V$, $b_1$, $b_2$, ..., $b_{m+C}$, and then perform one update step.

W.L.O.G. we assume the local client is client 1. We define the local loss function as follows:

$$
\mathcal{L}_L(V, B) = \mathbb{E}_{x \sim \mathcal{D}_1} \left[ \frac{1}{2} (V^T B x - V_1^{*T} B_* x)^2 \right],
$$

where $\mathcal{D}_1$ is the data distribution for client 1.

Now, let $(\mathbf{X}_1, \mathbf{Y}_1)$ represent the local dataset of client 1, consisting of $n_1$ data points $\{(x_{1j}, y_{1j})\}_{j=1}^{n_1}$. We aim to calculate the gradient of the empirical loss function with respect to the parameters. The empirical loss function is given by:

$$
\widehat{\mathcal{L}_L}(V, B) = \frac{1}{n_1} \sum_{j=1}^{n_1} \left[ \frac{1}{2} (V^T B x_{1j} - V_1^{*T} B_* x_{1j})^2 \right].
$$

In practice, we take the gradient of this empirical loss function with respect to the parameters $V$, $b_1$, $b_2$, ..., $b_{m+C}$. However, since we are particularly interested in computing the expectation $\mathbb{E}[b_j^{FT}]$, we evaluate the expected value of the gradients using one pass through the whole dataset as follows:

$$
\mathbb{E}\left[ \left. \frac{\partial \widehat{\mathcal{L}_L}}{\partial V} \right|_{\substack{V=V_0 \\ B=B_*}} \right] = \mathbb{E}\left[ \left. \frac{\partial}{\partial V} \left( \frac{1}{n_1} \sum_{j=1}^{n_1} \frac{1}{2} \left( V^T B x_{1j} - V_1^{*T} B_* x_{1j} \right)^2 \right) \right|_{\substack{V=V_0 \\ B=B_*}} \right]
$$

$$
= \frac{1}{n_1} \sum_{j=1}^{n_1} \mathbb{E}\left[ (V_0^T B_* x_{1j} - V_1^{*T} B_* x_{1j}) x_{1j}^T B_*^T \right]
$$

$$
= \mathbb{E}_{x \sim \mathcal{D}_1} \left[ (V_0^T B_* x - V_1^{*T} B_* x) x^T B_*^T \right].
$$

Therefore, let $V_0 = \begin{bmatrix} V_{com}^{*}{}^T & \mathbf{0}^T \end{bmatrix}^T$. It follows that:

$$
\mathbb{E}\left[ \left. \frac{\partial L}{\partial V} \right|_{\substack{V=V_0 \\ B=B_*}} \right] = \mathbb{E}_{x \sim \mathcal{D}_1} \left[ (V_0^T B_* x - V_1^{*T} B_* x) x^T B_*^T \right]
$$

$$
= \mathbb{E}_{x \sim \mathcal{D}_1} \left[ ((V_0 - V_1^*)^T B_* x) x^T B_*^T \right]
$$

$$
= (V_0 - V_1^*)^T B_* \left( \mathbb{E}_{x \sim \mathcal{D}_1} \left[ x x^T \right] \right) B_*^T
$$

$$
= (V_0 - V_1^*)^T B_* B_*^T \qquad \text{(second moment is identity)}
$$

$$
= (V_0 - V_1^*)^T. \qquad \text{($B_*$ has orthonormal rows)}
$$

Let $B_* = \begin{bmatrix} b_1^{*T} \\ \vdots \\ b_m^{*}{}^T \\ b_{m+1}^{*}{}^T \\ \vdots \\ b_{m+C}^{*}{}^T \end{bmatrix}$. Then, similarly, it can be shown that:

$$\mathbb{E}\left[\frac{\partial L}{\partial b_j}\bigg|_{\substack{V=V_0 \\ B=B_*}}\right] = \mathbb{E}_{x\sim\mathcal{D}_1}\left[(V_0^T B_* x - V_1^{*T} B_* x)(V_0)_j x^T\right]$$

$$= \mathbb{E}_{x\sim\mathcal{D}_1}\left[(V_0)_j\left((V_0 - V_1^*)^T B_* x\right)x^T\right]$$

$$= (V_0)_j (V_0 - V_1^*)^T B_*\left(\mathbb{E}_{x\sim\mathcal{D}_1}\left[x x^T\right]\right)$$

$$= (V_0)_j (V_0 - V_1^*)^T B_*. \qquad \text{(second moment is identity)}$$

Here, $(V_0)_j$ is the $j$-th element of the vector $V_0$. For learning rate $\eta$, one step of gradient descent is:

$$V_{FT} = V_0 - \eta\left(\frac{\partial L}{\partial V}\bigg|_{\substack{V=V_0 \\ B=B_*}}\right)^T$$

$$b_j^{FT} = b_j^* - \eta\left(\frac{\partial L}{\partial b_j}\bigg|_{\substack{V=V_0 \\ B_0=B_*}}\right)^T.$$

These two equations can be further refined as:

$$\mathbb{E}\left[V_{FT}\right] = \begin{bmatrix} V_{com}^{*T} & \mathbf{0}^T \end{bmatrix}^T - \eta(V_0 - V_1^*) = \begin{bmatrix} V_{com}^{*T} & \eta\lambda e_1^T \end{bmatrix}^T$$

$$\mathbb{E}\left[b_j^{FT}\right] = b_j^* - \eta\left(\frac{\partial L}{\partial b_j}\bigg|_{\substack{V=V_0 \\ B_0=B_*}}\right)^T = b_j^* - \eta\left((V_0)_j (V_0 - V_1^*)^T B_*\right)^T$$

$$= b_j^* - \eta\lambda\left((V_0)_j \begin{bmatrix} \mathbf{0}^T & -e_1^T \end{bmatrix} B_*\right)^T = b_j^* + \eta\lambda(V_0)_j b_{m+1}^*.$$

Therefore, we have:

$$\mathbb{E}\left[B_{FT}\right] = \begin{bmatrix} b_1^{*T} + \eta\lambda(V_0)_1 b_{m+1}^{*T} \\ \vdots \\ b_m^{*T} + \eta\lambda(V_0)_m b_{m+1}^{*T} \\ b_{m+1}^{*T} + \eta\lambda(V_0)_{m+1} b_{m+1}^{*T} \\ \vdots \\ b_{m+C}^{*T} + \eta\lambda(V_0)_{m+C} b_{m+1}^{*T} \end{bmatrix} = \begin{bmatrix} b_1^{*T} + \eta\lambda(V_0)_1 b_{m+1}^{*T} \\ \vdots \\ b_m^{*T} + \eta\lambda(V_0)_m b_{m+1}^{*T} \\ b_{m+1}^{*T} \\ \vdots \\ b_{m+C}^{*T} \end{bmatrix}$$

$$= \begin{bmatrix} b_1^{*T} + \eta\lambda(V_{com}^*)_1 b_{m+1}^{*T} \\ \vdots \\ b_m^{*T} + \eta\lambda(V_{com}^*)_m b_{m+1}^{*T} \\ b_{m+1}^{*T} \\ \vdots \\ b_{m+C}^{*T} \end{bmatrix}.$$

Similarly, if the fine-tuning is done over the data of client $i$, we would have:

$$\mathbb{E}\left[b_j^{FT}\right] = b_j^* + \eta\lambda(V_0)_j b_{m+i}^*,$$

which concludes the proof. $\qquad\qquad\qquad\square$

*Proof of Theorem 4.4.* We assume that the pre-trained model perfectly captures the feature extractor matrix $B_*$, and its linear head represents the common part shared across all clients, excluding any client-specific components of the ground-truth function. Thus, $B_0 = B_*$ and $V_0 = \begin{bmatrix} V_{com}^{*T} & \mathbf{0} \end{bmatrix}^T$.

In this setting, we analyze the effects of LP-FT and FT on the model parameters. For both LP-FT and FT, we determine the parameters after fine-tuning, compute the global loss, and then compare these global losses.

W.L.O.G. we assume that we are doing the fine-tuning over the local data of client 1. First, we study LP-FT. Initially, one step of linear probing is conducted with the fixed feature extractor $B_*$. After this step, the linear head $V_{LP}$ will converge to $V_1^*$. This is because we know that:

$$\arg\min_v \left\| \mathbf{X} B_0^\top v - \mathbf{X} B_*^\top v_* \right\|_2^2 = \left( B_0 \mathbf{X}^\top \mathbf{X} B_0^\top \right)^{-1} B_0 \mathbf{X}^\top \mathbf{X} B_*^\top v_*,$$

where $\mathbf{X}$ is the $n \times d$ matrix including data of $n$ individuals. Since the fine-tuning is on the data of the client 1 (local data), we have:

$$V_{LP} = \left( B_0 \mathbf{X_1}^\top \mathbf{X_1} B_0^\top \right)^{-1} B_0 \mathbf{X_1}^\top \mathbf{X_1} B_*^\top V_1^*.$$

Therefore, we have:

$$\begin{aligned}
V_{LP} &= \left( B_0 \mathbf{X_1}^\top \mathbf{X_1} B_0^\top \right)^{-1} B_0 \mathbf{X_1}^\top \mathbf{X_1} B_*^\top V_1^* \\
&= \left( B_* \mathbf{X_1}^\top \mathbf{X_1} B_*^\top \right)^{-1} B_* \mathbf{X_1}^\top \mathbf{X_1} B_*^\top V_1^* \\
&= V_1^*.
\end{aligned}$$

Since at the beginning of the fine-tuning (FT) step, we have the perfect $B_*$ and $V_1^*$ for the local client 1, and FT is performed on the data of the same client, we can conclude that after one step of FT following LP, the parameters will remain unchanged. Specifically, we have $V_{LPFT} = V_1^* = \begin{bmatrix} V_{com}^* {}^T & e_1^T \end{bmatrix}^T$ and $B_{LPFT} = B_*$.

For the performance on the global data, we have:

$$\begin{aligned}
\mathcal{L}_G(V_{LPFT}, B_{LPFT}) &= \frac{1}{C} \sum_{i \in [C]} \mathbb{E}_{x \sim \mathcal{D}_i} \left[ \frac{1}{2} (V_{LPFT}^T B_{LPFT} x - V_i^{*T} B_* x)^2 \right] \\
&= \frac{1}{C} \sum_{i \in [C]} \mathbb{E}_{x \sim \mathcal{D}_i} \left[ \frac{1}{2} (V_1^{*T} B_* x - V_i^{*T} B_* x)^2 \right] \\
&= \frac{1}{2C} \sum_{i \in [C]} \mathbb{E}_{x \sim \mathcal{D}_i} \left[ (B_*^T V_1^* - B_*^T V_i^*)^T x x^T (B_*^T V_1^* - B_*^T V_i^*) \right] \\
&= \frac{1}{2C} \sum_{i \in [C]} \left[ (B_*^T V_1^* - B_*^T V_i^*)^T \mathbb{E}_{x \sim \mathcal{D}_i} \left[ x x^T \right] (B_*^T V_1^* - B_*^T V_i^*) \right] \\
&= \frac{1}{2C} \sum_{i \in [C]} \left[ (B_*^T V_1^* - B_*^T V_i^*)^T (B_*^T V_1^* - B_*^T V_i^*) \right] \quad \text{(second moment } I_d) \\
&= \frac{1}{2C} \sum_{i \in [C]} \left[ (V_1^* - V_i^*)^T B_* B_*^T (V_1^* - V_i^*) \right] \\
&= \frac{1}{2C} \sum_{i \in [C]} \left[ (V_1^* - V_i^*)^T I_k (V_1^* - V_i^*) \right] \quad \text{(} B_* \text{ has orthonormal rows)} \\
&= \frac{1}{2C} \sum_{\substack{i \in [C] \\ i \neq 1}} \left[ (V_1^* - V_i^*)^T (V_1^* - V_i^*) \right] \\
&= \frac{1}{2C} \sum_{\substack{i \in [C] \\ i \neq 1}} \left[ \left\| (V_1^* - V_i^*) \right\|_2^2 \right]
\end{aligned}$$

$$= \frac{1}{2C} \sum_{\substack{i \in [C] \\ i \neq 1}} \left[ \left\| \left( [{V_{com}^*}^T \quad \lambda e_1^T]^T - [{V_{com}^*}^T \quad \lambda e_i^T]^T \right) \right\|_2^2 \right]$$

$$= (\frac{1}{2C})2(C-1) = \lambda^2 \frac{C-1}{C}. \tag{1}$$

It can be shown that:

$$\mathcal{L}_G(V_{FT}, B_{FT}) = \frac{1}{C} \sum_{i \in [C]} \mathbb{E}_{x \sim \mathcal{D}_i} \left[ \frac{1}{2}(V_{FT}^T B_{FT} x - {V_i^*}^T B_* x)^2 \right]$$

$$= \frac{1}{2C} \sum_{i \in [C]} \mathbb{E}_{x \sim \mathcal{D}_i} \left[ (B_{FT}^T V_{FT} - B_*^T V_i^*)^T x x^T (B_{FT}^T V_{FT} - B_*^T V_i^*) \right]$$

$$= \frac{1}{2C} \sum_{i \in [C]} (B_{FT}^T V_{FT} - B_*^T V_i^*)^T \left[ \mathbb{E}_{x \sim \mathcal{D}_i} x x^T \right] (B_{FT}^T V_{FT} - B_*^T V_i^*)$$

$$= \frac{1}{2C} \sum_{i \in [C]} (B_{FT}^T V_{FT} - B_*^T V_i^*)^T (B_{FT}^T V_{FT} - B_*^T V_i^*) \qquad \text{(second moment is } I_d\text{)}$$

$$= \frac{1}{2C} \sum_{i \in [C]} \left\| (B_{FT}^T V_{FT} - B_*^T V_i^*) \right\|_2^2. \tag{2}$$

We have:

$$B_*^T V_i^* = \sum_{j=1}^m (V_{com}^*)_j b_j^* + \lambda b_{m+i}^*$$

$$B_{FT}^T V_{FT} = \sum_{j=1}^m (V_{com}^*)_j b_j^* + \sum_{j=1}^m \eta\lambda(V_{com}^*)_j^2 b_{m+1}^* + \eta\lambda b_{m+1}^*.$$

Therefore, we can obtain:

$$(B_{FT}^T V_{FT} - B_*^T V_i^*) = \lambda\Big( \sum_{j=1}^m \eta(V_{com}^*)_j^2 b_{m+1}^* + \eta b_{m+1}^* - b_{m+i}^* \Big).$$

For $i \neq 1$, we have:

$$(B_{FT}^T V_{FT} - B_*^T V_i^*)^T (B_{FT}^T V_{FT} - B_*^T V_i^*)$$

$$= \lambda^2 (\sum_{j=1}^m \eta(V_{com}^*)_j^2 b_{m+1}^* + \eta b_{m+1}^* - b_{m+i}^*)^T (\sum_{j=1}^m \eta(V_{com}^*)_j^2 b_{m+1}^* + \eta b_{m+1}^* - b_{m+i}^*)$$

$$= \lambda^2 \left( \Big( \eta + \eta\sum_{j=1}^m (V_{com}^*)_j^2 \Big)^2 + 1 \right). \qquad \text{(rows of } B_* \text{ are orthonormal)}$$

For $i = 1$, we have:

$$(B_{FT}^T V_{FT} - B_*^T V_i^*)^T (B_{FT}^T V_{FT} - B_*^T V_i^*)$$

$$= \lambda^2 (\sum_{j=1}^m \eta(V_{com}^*)_j^2 b_{m+1}^* + \eta b_{m+1}^* - b_{m+i}^*)^T (\sum_{j=1}^m \eta(V_{com}^*)_j^2 b_{m+1}^* + \eta b_{m+1}^* - b_{m+i}^*)$$

$$= \lambda^2 \Big( \eta + \eta\sum_{j=1}^m (V_{com}^*)_j^2 - 1 \Big)^2. \qquad \text{(rows of } B_* \text{ are orthonormal)}$$

Combining these with (2), we can conclude:

$$\mathcal{L}_G(V_{FT}, B_{FT}) = \frac{1}{2C} \sum_{i \in [C]} \left\| (B_{FT}^T V_{FT} - B_*^T V_i^*) \right\|_2^2$$

$$= \frac{\lambda^2}{2C} \left( \left( \eta + \eta \sum_{j=1}^m (V_{com}^*)_j^2 - 1 \right)^2 + (C-1) \left( \left( \eta + \eta \sum_{j=1}^m (V_{com}^*)_j^2 \right)^2 + 1 \right) \right).$$

(3)

Combining (1) and (3), we have:

$$\mathcal{L}_G(V_{LPFT}, B_{LPFT}) \le \mathcal{L}_G(V_{FT}, B_{FT}).$$

$\square$

*Proof of Theorem 4.5.* W.L.O.G. we assume that the local fine-tuning is performed on the data of the first client. Initially, one step of linear probing is conducted with the fixed feature extractor $B_*$. After this step, the linear head $V_{LP}$ will converge to $V_1^*$. This is because we know that:

$$\arg\min_v \left\| \mathbf{X_1} B_0^\top v - \mathbf{X_1} B_*^\top v_* \right\|_2^2 = \left( B_0 \mathbf{X_1}^\top \mathbf{X_1} B_0^\top \right)^{-1} B_0 \mathbf{X_1}^\top \mathbf{X_1} B_*^\top v_*,$$

where $\mathbf{X_1}$ is the $n \times d$ matrix including data of $n$ individuals. Since the fine-tuning is on the data of the client 1 (local data), we have:

$$V_{LP} = \left( B_0 \mathbf{X_1}^\top \mathbf{X_1} B_0^\top \right)^{-1} B_0 \mathbf{X_1}^\top \mathbf{X_1} B_*^\top V_1^*.$$

Therefore, we have:

$$V_{LP} = \left( B_0 \mathbf{X_1}^\top \mathbf{X_1} B_0^\top \right)^{-1} B_0 \mathbf{X_1}^\top \mathbf{X_1} B_*^\top V_1^*$$

$$= \left( B_* \mathbf{X_1}^\top \mathbf{X_1} B_*^\top \right)^{-1} B_* \mathbf{X_1}^\top \mathbf{X_1} B_*^\top V_1^*$$

$$= V_1^*.$$

This part is identical to the initial part of the proof of Theorem 4.4. Since at the beginning of the fine-tuning (FT) step, we have the perfect $B_*$ and $V_1^*$ for the local client 1, and FT is performed on the data of the same client, we can conclude that after one step of FT following LP, the parameters will remain unchanged. Specifically, we have $V_{LPFT} = V_1^* = \begin{bmatrix} V_{com}^* {}^T & \lambda e_1^T \end{bmatrix}^T$ and $B_{LPFT} = B_*$.

For the performance on the global data, we have:

$$\mathcal{L}_G(V_{LPFT}, B_{LPFT}) = \frac{1}{C} \sum_{i \in [C]} \mathbb{E}_{x \sim \mathcal{D}_i} \left[ \frac{1}{2} (V_{LPFT}^T B_{LPFT} x - V_i^* {}^T B_* x)^2 \right]$$

$$= \frac{1}{C} \sum_{i \in [C]} \mathbb{E}_{x \sim \mathcal{D}_i} \left[ \frac{1}{2} (V_1^* {}^T B_* x - V_i^* {}^T B_* x)^2 \right]$$

$$= \frac{1}{2C} \sum_{i \in [C]} \mathbb{E}_{x \sim \mathcal{D}_i} \left[ (B_*^T V_1^* - B_*^T V_i^*)^T x x^T (B_*^T V_1^* - B_*^T V_i^*) \right]$$

$$= \frac{1}{2C} \sum_{i \in [C]} \left[ (B_*^T V_1^* - B_*^T V_i^*)^T \mathbb{E}_{x \sim \mathcal{D}_i} \left[ x x^T \right] (B_*^T V_1^* - B_*^T V_i^*) \right]$$

$$= \frac{1}{2C} \sum_{i \in [C]} \left[ (V_1^* - V_i^*)^T B_* \left( \mathbb{E}_{x \sim \mathcal{D}_i} \left[ x x^T \right] \right) B_*^T (V_1^* - V_i^*) \right]$$

$$= \frac{1}{2C} \sum_{i \in [C]} \left[ (V_1^* - V_i^*)^T B_* \left( \mathbb{E}_{n \sim \mathcal{N}(0, I_d)} \left[ (e_i + \epsilon n)(e_i + \epsilon n)^T \right] \right) B_*^T (V_1^* - V_i^*) \right]$$

$$= \frac{1}{2C} \sum_{i \in [C]} \left[ (V_1^* - V_i^*)^T B_* \left( e_i e_i^T + \epsilon^2 \mathbb{E}_{n \sim \mathcal{N}(0, I_d)} \left[ nn^T \right] \right) B_*^T (V_1^* - V_i^*) \right]$$

$$= \frac{1}{2C} \sum_{i \in [C]} \left[ (V_1^* - V_i^*)^T B_* \left( e_i e_i^T + \epsilon^2 I_d \right) B_*^T (V_1^* - V_i^*) \right]$$

$$= \frac{1}{2C} \sum_{i \in [C]} \left[ (V_1^* - V_i^*)^T B_* \left( e_i e_i^T \right) B_*^T (V_1^* - V_i^*) \right]$$

$$+ \frac{1}{2C} \sum_{i \in [C]} \left[ (V_1^* - V_i^*)^T B_* \left( \epsilon^2 I_d \right) B_*^T (V_1^* - V_i^*) \right]$$

$$= \frac{1}{2C} \sum_{i \in [C]} \left[ (V_1^* - V_i^*)^T (B_*)_{:,i} (B_*)_{:,i}^T (V_1^* - V_i^*) \right] \qquad ((B_*)_{:,i} \ i\text{-th column of } B_*)$$

$$+ \frac{1}{2C} \sum_{i \in [C]} \left[ \epsilon^2 (V_1^* - V_i^*)^T (V_1^* - V_i^*) \right] \qquad\qquad (B_* \text{ has orthonormal rows})$$

$$= \frac{\lambda^2}{2C} \sum_{i \in [C]} \left[ \left( (B_*)_{m+1,i} - (B_*)_{m+i,i} \right)^2 \right] + \frac{1}{2C} \epsilon^2 \sum_{i \in [C]} \left[ \left\| (V_1^* - V_i^*) \right\|_2^2 \right]$$

$$= \frac{\lambda^2}{2C} \sum_{i \in [C]} \left[ \left( (B_*)_{m+1,i} - (B_*)_{m+i,i} \right)^2 \right] + \frac{\lambda^2 (C-1)}{C} \epsilon^2. \qquad (4)$$

We want to analyze the fine tuning (FT) method, focusing on the effect of initial parameters. We perform one pass through the entire dataset to simulate the complete fine-tuning process. Consider the Mean Squared Error (MSE) loss function with parameters $V$ and $B$, where $B$ is represented as follows:

$$B = \begin{bmatrix} b_1^T \\ \vdots \\ b_m^T \\ b_{m+1}^T \\ \vdots \\ b_{m+C}^T \end{bmatrix},$$

where $b_i^T \in \mathbb{R}^{1 \times d}$ denotes the $i$-th row of matrix $B$, and $m + C = k$.

To apply one step of gradient descent, we need to compute the gradient of the loss function with respect to $V, b_1, b_2, \ldots, b_{m+C}$, and then perform one update step.

Let $V_0 = \begin{bmatrix} V_{com}^{*}{}^T & \mathbf{0}^T \end{bmatrix}^T$. It follows that:

$$\mathbb{E} \left[ \frac{\partial L}{\partial V} \bigg|_{\substack{V=V_0 \\ B=B_*}} \right] = \mathbb{E}_{x \sim \mathcal{D}_1} \left[ (V_0^T B_* x - V_1^{*T} B_* x) x^T B_*^T \right]$$

$$= \mathbb{E}_{x \sim \mathcal{D}_1} \left[ \left( (V_0 - V_1^*)^T B_* x \right) x^T B_*^T \right]$$

$$= (V_0 - V_1^*)^T B_* \left( \mathbb{E}_{x \sim \mathcal{D}_1} \left[ xx^T \right] \right) B_*^T$$

$$= (V_0 - V_1^*)^T B_* \left( \mathbb{E}_{n \sim \mathcal{N}(0, I_d)} \left[ (e_1 + \epsilon n)(e_1 + \epsilon n)^T \right] \right) B_*^T$$

$$= (V_0 - V_1^*)^T B_* \left( e_1 e_1^T + \epsilon^2 I_d \right) B_*^T$$

$$= (V_0 - V_1^*)^T B_* \left( e_1 e_1^T \right) B_*^T + (V_0 - V_1^*)^T B_* \left( \epsilon^2 I_d \right) B_*^T$$

$$= (V_0 - V_1^*)^T B_* \left( e_1 e_1^T \right) B_*^T + \epsilon^2 (V_0 - V_1^*)^T \qquad (B_* \text{ has orthonormal rows})$$

$$= (V_0 - V_1^*)^T \left( (B_*)_{:,1} (B_*)_{:,1}^T \right) + \epsilon^2 (V_0 - V_1^*)^T \qquad ((B_*)_{:,1} \text{ is first column of } B_*)$$

$$= \left( -\lambda (B_*)_{m+1,1} \right) (B_*)_{:,1}^T + \epsilon^2 (V_0 - V_1^*)^T.$$

Let $B_* = \begin{bmatrix} b_1^{*T} \\ \vdots \\ b_m^{*T} \\ b_{m+1}^{*T} \\ \vdots \\ b_{m+C}^{*T} \end{bmatrix}$. Then, it can be shown that:

$$\mathbb{E}\left[ \frac{\partial L}{\partial b_j} \bigg|_{\substack{V=V_0 \\ B=B_*}} \right] = \mathbb{E}_{x \sim \mathcal{D}_1} \left[ (V_0^T B_* x - V_1^{*T} B_* x)(V_0)_j x^T \right]$$

$$= \mathbb{E}_{x \sim \mathcal{D}_1} \left[ (V_0)_j \left( (V_0 - V_1^*)^T B_* x \right) x^T \right]$$

$$= (V_0)_j (V_0 - V_1^*)^T B_* \left( \mathbb{E}_{x \sim \mathcal{D}_1} \left[ xx^T \right] \right)$$

$$= (V_0)_j (V_0 - V_1^*)^T B_* \left( \mathbb{E}_{n \sim \mathcal{N}(0, I_d)} \left[ (e_1 + \epsilon n)(e_1 + \epsilon n)^T \right] \right)$$

$$= (V_0)_j (V_0 - V_1^*)^T B_* \left( e_1 e_1^T + \epsilon^2 I_d \right)$$

$$= (V_0)_j (V_0 - V_1^*)^T B_* \left( e_1 e_1^T \right) + (V_0)_j (V_0 - V_1^*)^T B_* \left( \epsilon^2 I_d \right)$$

$$= (V_0)_j (V_0 - V_1^*)^T B_* \left( e_1 e_1^T \right) + \epsilon^2 (V_0)_j (V_0 - V_1^*)^T B_*.$$

Here, $(V_0)_j$ is the $j$-th element of the vector $V_0$. For learning rate $\eta$, one step of gradient descent can be:

$$V_{FT} = V_0 - \eta \left( \frac{\partial L}{\partial V} \bigg|_{\substack{V=V_0 \\ B=B_*}} \right)^T$$

$$b_j^{FT} = b_j^* - \eta \left( \frac{\partial L}{\partial b_j} \bigg|_{\substack{V=V_0 \\ B_0=B_*}} \right)^T.$$

These two equations can be further refined as:

$$\mathbb{E}\left[ V_{FT} \right] = \begin{bmatrix} V_{com}^{*T} & \mathbf{0}^T \end{bmatrix}^T - \eta \left( -\lambda (B_*)_{m+1,1} (B_*)_{:,1} + \epsilon^2 (V_0 - V_1^*) \right)$$

$$= \begin{bmatrix} V_{com}^* \\ 0 \\ 0 \\ \vdots \\ 0 \end{bmatrix} + \begin{bmatrix} \mathbf{0} \\ \eta \lambda \epsilon^2 \\ 0 \\ \vdots \\ 0 \end{bmatrix} + \eta \lambda (B_*)_{m+1,1} (B_*)_{:,1}$$

$$\mathbb{E}\left[ b_j^{FT} \right] = b_j^* - \eta \left( (V_0)_j (V_0 - V_1^*)^T B_* \left( e_1 e_1^T \right) + \epsilon^2 (V_0)_j (V_0 - V_1^*)^T B_* \right)^T$$

$$= b_j^* + \begin{bmatrix} \eta \lambda (V_0)_j (B_*)_{m+1,1} \\ 0 \\ \vdots \\ 0 \end{bmatrix} + \eta \lambda (V_0)_j \epsilon^2 b_{m+1}^*.$$

Therefore, we have:

$$\mathbb{E}\left[B_{FT}\right] = \begin{bmatrix} b_1^{*T} + \eta\lambda(V_0)_1\epsilon^2 {b_{m+1}^*}^T + \eta\lambda(V_0)_1(B_*)_{m+1,1}e_1^T \\ \vdots \\ {b_m^*}^T + \eta\lambda(V_0)_m\epsilon^2 {b_{m+1}^*}^T + \eta\lambda(V_0)_m(B_*)_{m+1,1}e_1^T \\ {b_{m+1}^*}^T + \eta\lambda(V_0)_{m+1}\epsilon^2 {b_{m+1}^*}^T + \eta\lambda(V_0)_{m+1}(B_*)_{m+1,1}e_1^T \\ \vdots \\ {b_{m+C}^*}^T + \eta\lambda(V_0)_{m+C}\epsilon^2 {b_{m+1}^*}^T + \eta\lambda(V_0)_{m+C}(B_*)_{m+1,1}e_1^T \end{bmatrix}$$

$$= \begin{bmatrix} b_1^{*T} + \eta\lambda(V_0)_1\epsilon^2 {b_{m+1}^*}^T + \eta\lambda(V_0)_1(B_*)_{m+1,1}e_1^T \\ \vdots \\ {b_m^*}^T + \eta\lambda(V_0)_m\epsilon^2 {b_{m+1}^*}^T + \eta\lambda(V_0)_m(B_*)_{m+1,1}e_1^T \\ {b_{m+1}^*}^T \\ \vdots \\ {b_{m+C}^*}^T \end{bmatrix}.$$

It can be shown that:

$$\mathcal{L}_G(V_{FT}, B_{FT}) = \frac{1}{C}\sum_{i\in[C]}\mathbb{E}_{x\sim\mathcal{D}_i}\left[\frac{1}{2}(V_{FT}^T B_{FT}x - V_i^{*T}B_*x)^2\right]$$

$$= \frac{1}{2C}\sum_{i\in[C]}\mathbb{E}_{x\sim\mathcal{D}_i}\left[(B_{FT}^T V_{FT} - B_*^T V_i^*)^T xx^T(B_{FT}^T V_{FT} - B_*^T V_i^*)\right]$$

$$= \frac{1}{2C}\sum_{i\in[C]}(B_{FT}^T V_{FT} - B_*^T V_i^*)^T\left[\mathbb{E}_{x\sim\mathcal{D}_i}xx^T\right](B_{FT}^T V_{FT} - B_*^T V_i^*)$$

$$= \frac{1}{2C}\sum_{i\in[C]}(B_{FT}^T V_{FT} - B_*^T V_i^*)^T\left(e_ie_i^T + \epsilon^2 I_d\right)(B_{FT}^T V_{FT} - B_*^T V_i^*)$$

$$= \frac{1}{2C}\sum_{i\in[C]}(B_{FT}^T V_{FT} - B_*^T V_i^*)^T\left(e_ie_i^T\right)(B_{FT}^T V_{FT} - B_*^T V_i^*)$$

$$+ \frac{1}{2C}\sum_{i\in[C]}(B_{FT}^T V_{FT} - B_*^T V_i^*)^T\left(\epsilon^2 I_d\right)(B_{FT}^T V_{FT} - B_*^T V_i^*)$$

$$= \frac{1}{2C}\sum_{i\in[C]}(B_{FT}^T V_{FT} - B_*^T V_i^*)^T\left(e_ie_i^T\right)(B_{FT}^T V_{FT} - B_*^T V_i^*)$$

$$+ \epsilon^2\frac{1}{2C}\sum_{i\in[C]}\left\|(B_{FT}^T V_{FT} - B_*^T V_i^*)\right\|_2^2. \tag{5}$$

We have:

$$B_*^T V_i^* = \sum_{j=1}^m (V_{com}^*)_j b_j^* + \lambda b_{m+i}^*$$

$$B_{FT}^T V_{FT} = \eta\lambda\epsilon^2\sigma^2 b_{m+1}^* + \sum_{j=m+1}^{m+C}\left(\eta\lambda(B_*)_{m+1,1}(B_*)_{j,1}\right)b_j^*$$

$$+ \sum_{j=1}^m\left((V_{com}^*)_j + \eta\lambda(B_*)_{m+1,1}(B_*)_{j,1}\right)\left(b_j^* + \eta\lambda(V_{com}^*)_j\epsilon^2\sigma^2 b_{m+1}^* + \eta\lambda(V_{com}^*)_j(B_*)_{m+1,1}e_1\right).$$

Therefore, we can obtain:

$$B_{FT}^T V_{FT} - B_*^T V_i^* = \sum_{j=1}^{m} (V_{com}^*)_j \left( b_j^* + \eta\lambda(V_{com}^*)_j \epsilon^2 \sigma^2 b_{m+1}^* + \eta\lambda(V_{com}^*)_j (B_*)_{m+1,1} e_1 \right)$$

$$+ \eta\lambda\epsilon^2\sigma^2 b_{m+1}^* - \lambda b_{m+i}^* + \sum_{j=m+1}^{m+C} \left( \eta\lambda(B_*)_{m+1,1}(B_*)_{j,1} \right) b_j^* \tag{6}$$

$$+ \sum_{j=1}^{m} \left( \eta\lambda(B_*)_{m+1,1}(B_*)_{j,1} \right) \left( b_j^* + \eta\lambda(V_{com}^*)_j \epsilon^2 \sigma^2 b_{m+1}^* + \eta\lambda(V_{com}^*)_j (B_*)_{m+1,1} e_1 \right). \tag{7}$$

From equation (4), we observe that $\mathcal{L}_G(V_{LPFT}, B_{LPFT})$ is a monotonically decreasing function of $\lambda$ and as $\lambda$ approaches zero, $\mathcal{L}_G(V_{LPFT}, B_{LPFT})$ also converges to zero. In contrast, combining equations (5) and (6), we find that $\mathcal{L}_G(V_{FT}, B_{FT})$ does not converge to zero as $\lambda$ approaches zero due to the presence of constant terms independent of $\lambda$. Therefore, it follows that there always exists a threshold $\lambda^*$ such that for all $\lambda \le \lambda^*$:

$$\mathcal{L}_G(V_{LPFT}, B_{LPFT}) \le \mathcal{L}_G(V_{FT}, B_{FT}).$$

□

## G EMPIRICAL PERFORMANCE OF LP-FT AND FT UNDER THEOREM 4.5 CONDITIONS

To give a better understanding of Theorem 4.5, we give a simple visualization of two randomly generated data-generating functions for different clients and compute the global loss of LP-FT and FT based on equations (4) and (5).

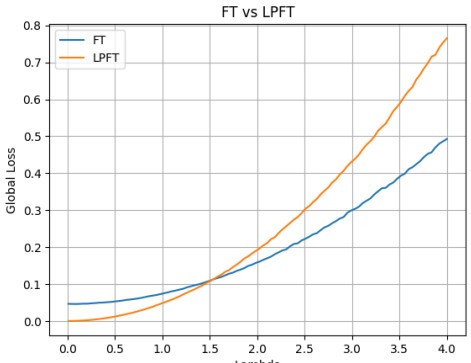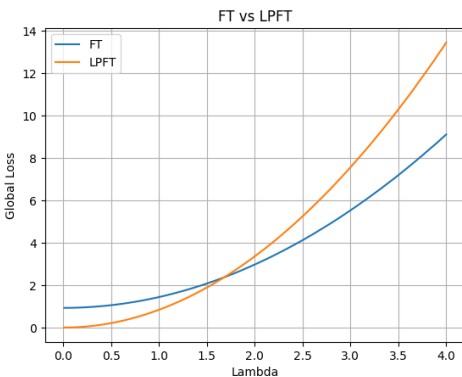

Figure 6: (a) Global loss of LP-FT and FT as a function of the heterogeneity parameter $\lambda$, with $\eta = 0.1$, $\epsilon = 0.1$, matrix $B_*$ as a $10 \times 20$ random matrix, and number of clients $C = 5$. (b) Global loss of LP-FT and FT as a function of the heterogeneity parameter $\lambda$, with $\eta = 0.1$, $\epsilon = 1$, matrix $B_*$ as a $10 \times 20$ random matrix, and number of clients $C = 5$.

These examples illustrate, within the theoretical setting of Sec. 4.2, the behavior of the loss functions for LP-FT and FT with a randomly generated labeling function $y = V_i^{*T} B_* x$, a fixed learning rate $\eta$, noise parameter $\epsilon$, and a fixed number of clients $C$. To compute this, we generated 1000 random matrices $B_*$ and 1000 randomly chosen linear heads $V_i^*$ as ground-truth labeling functions, ensuring they adhere to the theoretical assumptions. Using equations (4) and (5), we calculated the average loss of LP-FT and FT across these random trials.

As shown in Fig. 6, there exists a threshold $\lambda^*$ such that when $\lambda \le \lambda^*$, LP-FT consistently outperforms FT. While this is a simplified example with a fixed number of clients, learning rate, noise parameter,

and dimensionality of the ground-truth parameters $B_*$ and $V_i^*$, the observed trend remains similar across different parameter settings. The purpose of this figure is to provide an intuitive understanding of Theorem 4.5 in a controlled, simplified context. More comprehensive experiments in Sec. 5 demonstrate that LP-FT globally outperforms FT across a broader range of heterogeneity levels in real-world settings.

## H    COMPUTATIONAL COST OF LP-FT COMPARED TO FT

In this section, we want to study the computation cost of adding one step of linear probing (LP) to the full-fine tuning (FT) to see how this additional LP step affects computational cost. Suppose the dimension of the output of the feature extractor layer (input of the linear head) is $d$ and the dimension of the output of the linear head is $m$. In fact, the linear head will be a $d \times m$ linear layer. We assume having $n$ samples. We want to see what is the computational cost of fine-tuning this linear head.

To estimate the computational cost of training a linear neural network layer with $d$ inputs, $m$ outputs, and $n$ samples, we analyze the steps involved:

1. **Forward Pass:** A linear neural network computes outputs as:

$$Y = XW,$$

   where:
   - $X \in \mathbb{R}^{n \times d}$ is the input matrix (with $n$ samples, each of dimension $d$),
   - $W \in \mathbb{R}^{d \times m}$ is the weight matrix,
   - $Y \in \mathbb{R}^{n \times m}$ is the output matrix.

   The cost of this matrix multiplication is $O(ndm)$.

2. **Backward Pass (Gradient Computation):** To update $W$, the gradient of the loss $\mathcal{L}$ with respect to $W$ is computed. We know that:

$$\nabla_W \mathcal{L} = X^T (\nabla_Y \mathcal{L})$$

   Therefore, computing $(\nabla_W \mathcal{L})$ involves:
   - Computing the gradient of the loss with respect to the outputs $Y$, which has a cost of $O(nk)$,
   - matrix multiplication $\nabla_W \mathcal{L} = X^T (\nabla_Y \mathcal{L})$ which involves a matrix multiplication with a cost of $O(ndm)$.

3. **Weight Update:** If using gradient descent, the cost of updating the weights is $O(dm)$.

**Total Computational Cost:** The total cost for one forward and backward pass through the data is dominated by $O(ndm)$, which accounts for both forward propagation and gradient computation. If the training involves multiple epochs, the total cost scales as:

$$O(e \cdot ndm),$$

where $e$ is the number of epochs.

