# OpenReview forum: "A Closer Look at Personalized Fine-Tuning in Heterogeneous Federated Learning"
_ICLR.cc/2025/Conference — Submitted to ICLR 2025_

### Official Review · Reviewer_sf8C · 2024-11-02

**Soundness:** 3
**Presentation:** 2
**Contribution:** 2
**Rating:** 6
**Confidence:** 3

**Summary:**

This paper proposes a fine-tuning-based personalized federated learning framework, aiming to solve the data heterogeneity challenge. Compared with existing methods that full fine-tuning the global, the paper combines the linear probing before full fine-tuning, which alleviates the local model overfitting issue. Besides, the paper conduct both experimental evaluation and theoretical analysis to demonstrate the method's effectiveness.

**Strengths:**

S1: The paper focuses on a long-standing and challenging issue in the FL, i.e., data heterogeneity.

S2: The proposed method is simple and easy to follow, and experimental evaluations show that it outperforms the baseline methods in multiple data heterogeneity scenarios.

**Weaknesses:**

W1: The technical contribution is limited. As the paper claims, the proposed LP-FT mechanism has been developed and explored in centralized setting, and the paper just adapts the method under the FL setting without more technical improvements.

W2: Some clarifications are confusing and hard to comprehend, please refer to section "Questions" for details.

**Questions:**

1. What is the exact meaning of LP-FT? There are no specific descriptions about the implementation details in the main paper.
2. Why is fine-tuning the linear head first and then followed by fine-tuning the full model better than direct full fine-tuning?
3. The theoretical analysis is hard to follow and it is recommended to give a concise summary about the analytical logic.
4. In Figure 2, the paper just illustrates the sum of global accuracy and local accuracy. However, it is recommended to supplement the individual performance to show that the local model's performance increases. From the intuitive view, the local model performance should decrease due to overfitting.

---

> ### Author Response · Authors · 2024-11-25
>
> We sincerely thank the reviewer for their valuable comments and questions. Below, we provide our responses to the concerns and inquiries raised:
>
> **Response to W1**
>
> We would like to clarify that the novelty and contribution of our work extend beyond simply adapting an existing method (LP-FT [1]) to a new scenario (PFT). Our key contributions lie in the **methodological and theoretical novelty**, as well as **empirical findings, benchmarking, and analysis**. These efforts provide **deeper insights** and **fine-grained evaluations** of diverse distribution shifts, demonstrating the effectiveness of LP-FT in the **new PFT setting**.
>
> **Methodological Novelties (See Section 1 Contribution Insight)**
>
> **1. Simple yet Non-Trivial Method: Novel Focus on Final-Stage LP-FT in FL**
>
> While our method may appear straightforward, its novelty lies in **strategically shifting the insertion phase** of LP-FT to the **final local training stage** in FL. Unlike most existing PFL methods that require modifications to the entire training process, including both General FL (GFL) training and local personalization, our approach focuses on the **critical yet underexplored final local training stage**. This makes our method a **versatile solution**, **independent of the GFL training process**. This shift not only introduces a **fresh perspective on FL personalization** but also **demonstrates significant theoretical and empirical insights**.
>
> We also respectfully wish reviewers to agree that **Simplicity does not diminish the novelty of our method**; instead, it enhances its practical impact and potential for widespread applicability in real-world FL scenarios. A simple yet effective method, like ours, is a strength that enables broader adoption while providing substantial improvements over state-of-the-art techniques.
>
> **2. Pioneering LP-FT to FL:**
>
> To the best of our knowledge, this work is **the first to systematically explore and leverage** the potential of the simple yet effective **LP-FT** method within the **FL** framework. This represents a significant contribution, as it opens up a new line of research exploring the application and adaptation of LP-FT in addressing the unique challenges of FL, such as data heterogeneity and distribution shifts.
>
> **Theoretical Novelties (See Section 1 Theory Contribution)**,
>
> **1. Our theoretical setting and assumption differ significantly from that of [1].**
> While [1] assumes a **centralized setting**, we focus on **personalized fine-tuning (PFT)** in FL, a fundamentally different scenario. In our case, the pre-trained model is trained on data from all clients, reflecting a global client distribution. This contrasts with the centralized setting in [1], where the pre-trained model is trained without any notion of a global client distribution.
>
> **2. Our theoretical analysis diverges significantly from that of [1].**
> In [1], the assumption is that a single ground-truth data-generating function underlies all data points. This assumption makes it challenging to study LP-FT in different distribution shift settings in PFT for FL (e.g., concept shift). In contrast, we assume that each client has a distinct ground-truth function, enabling the exploration of personalized fine-tuning in FL under varying distribution shifts. Specifically, the way we define the data-generating function (**Assumption 4.1**), with unique linear heads before pre-training $V_i^* = \begin{bmatrix} {V_{com}^*}^T & \lambda e_i^T \end{bmatrix}^T $ for each client and the way we define data heterogeneity $ \Bigl( x_i = e_i + \epsilon n, \text{ with } \( n \sim \mathcal{N}(0, I) \Bigr)$ , allow us to study both concept shift and covariate shift using gradient descent to compare FT and LP-FT in terms of global performance. This approach provides new insights into the challenges of PFT in FL. In other words, the main challenge we address in our theoretical analysis lies in incorporating assumptions that align the theoretical framework with our specific setup. Our analysis connects the superior global performance of the LP-FT method to the concept of feature distortion. This novel perspective offers valuable insights into the effectiveness of LP-FT in FL through the lens of feature distortion.
>
> **3. In-depth Integrating Theoretical Analysis and Empirical Studies:**
> We have conducted additional empirical studies to substantiate the theoretical analysis on the heterogeneity level $\lambda$. These include a comprehensive investigation with diverse ablation studies across varying levels of label-flipping ratios (**Section 5**). This **significantly extends the scope and granularity of distribution shifts** explored in the original LP-FT paper [1], providing deeper insights into its robustness and applicability under heterogeneous conditions.
>
> **[Continued below]**

---

> ### Author Response · Authors · 2024-11-25
>
> **[Continued from above]**
>
> **Response to W1**
>
> **Empirical Contributions (See Section 1 Contribution Evaluation):**,
>
> **1. New Observations: Detailed Analysis on Overfitting in PFT:**
> As shown in **Figure 2**, we analyze commonly used fine-tuning strategies and demonstrate their **tendency to overfit**, even with carefully tuned hyperparameters, across three different types of distribution shifts. These findings highlight a critical gap in prior FL research, where such tendencies **have not been systematically explored**.
>
> **2. Comprehensive Benchmarking PFT:**
> Our experiments span **seven diverse datasets** (CIFAR-10-C, CIFAR-10, Digit5, DomainNet, Chexpert, CelebA, CIFAR-10), addressing **four types of distribution shifts** (input-level shift, feature shift, spurious correlation, and label shift, as detailed in **Appendix Table 6**). We evaluate **five Personalized Fine-Tuning (PFT) methods** using **five robust metrics**, providing a thorough and comprehensive analysis.
>
> **3. Establishing New Ablation Studies to Support Theoretical Results:**
> In **Figure 4**, we conduct detailed analysis on feature distortion relation to overfitting and in **Table 2**, we conduct large ranges of label flipping to support the theoretical analysis validating our theoretical insights in low heterogeneity regimes and further demonstrating LP-FT’s superior performance in scenarios with higher heterogeneity.
>
>
>
> Reference
> - [1] Ananya Kumar, Aditi Raghunathan, Robbie Matthew Jones, Tengyu Ma, and Percy Liang. Fine-tuning can distort pretrained features and underperform out-of-distribution. In ICLR. OpenReview.net, 2022.

---

> ### Author Response · Authors · 2024-11-25
>
> **Response to Q1 & Q2**
>
> Thank you for the question. The superior performance of LP-FT (Linear-Probing-then-Fine-Tuning)—which involves fine-tuning the linear head followed by fine-tuning the full model (see **Section 1 Line 83**, and **Figure 1(b)**; further details are provided in **Appendix B.4** of the revised version)—can be explained both empirically and theoretically. Below, we first provide a high-level explanation and then point to the sections where these reasons are explored in depth.
>
> **1. High-Level Intuition**
> When applying LP-FT, the features (last-layer representations before the linear head) are fixed during the linear head fine-tuning (LP stage), while only the linear head is updated. In the subsequent full fine-tuning stage, since the linear head is already adapted to local performance, the features experience less distortion compared to directly applying FT without LP. This reduced feature distortion leads to better performance for LP-FT (stated in **Section 1 Line 82-90**). The pre-trained model already performs well on global data, so minimizing feature distortion ensures better global performance after fine-tuning. This concept is illustrated graphically in **Fig. 3**.
>
> **2. Empirical Evidence**
> In **Sec. 3.5**, we present experiments to explore the **connection** between federated **feature distortion** and the **performance of LP-FT and FT**. As shown in **Fig. 4(a)**, FT results in significant feature distortion, causing a substantial decline in global performance. In contrast, LP-FT maintains higher global performance with significantly reduced feature distortion.
>
> **3. Theoretical Insight**
> In **Sec. 4**, we provide a **theoretical analysis** comparing the global performance of LP-FT and FT under two distribution shift settings: concept shift and combined concept-and-covariate shifts. Using a two-layer neural network as the data-generating function and model structure, we simulate LP-FT and FT, where layer 1 represents the features and layer 2 represents the linear head.
> In **Lemma 4.3**, we derive the changes in features under FT inspired by feature distortion.
> In **Theorems 4.4 and 4.5**, we show that LP-FT achieves better global performance than FT under concept shift and combined concept-and-covariate shifts, respectively. These results demonstrate how reduced feature distortion in LP-FT contributes to its superior performance.
>
> **Summary**
> LP-FT outperforms FT in global performance by significantly reducing federated feature distortion, leading to improved overall results.

---

> ### Author Response · Authors · 2024-11-25
>
> **Response to Q3**
>
> We appreciate the suggestion to provide a concise summary of the theoretical analysis. We have included this **summary at the beginning of the theoretical analysis section** to offer a **clear road map** of the theory section.
>
> Here is the overview:
> To compare the global performance of LP-FT and FT, we make assumptions about the clients’ **data-generating function** (**Assumption 4.1**) and the **model structure** (**Assumption 4.2**). Both assumptions involve a two-layer network to distinguish between the linear head and the feature extractor layer. We then define the global loss function as MSE loss. Based on these assumptions, we compare the global performance of LP-FT and FT under **concept shift** (**Theorem 4.4**) and **combined concept-and-covariate shift** (**Theorem 4.5**). For this comparison, we use one step of gradient descent (a single pass through the entire dataset) to evaluate the global loss of both methods.
>
> ---
>
> **Response to Q4**
>
> Thank you for the suggestion. We have revised **Figure 2** to provide a detailed visualization of the individual global and local accuracy trends across various distribution shift scenarios. The updated figure includes:
> - (a) Global and local accuracy under different **learning rates** for full-parameter fine-tuning.
> - (b) Accuracy trends under varying **sparsity rates** for sparse fine-tuning.
> - (c) Accuracy behavior with different **regularization strengths** in proximal fine-tuning.
>
> In all subplots, global accuracy is represented by solid lines and local accuracy by dashed lines. The results show that while global accuracy remains stable, local accuracy consistently decreases across all hyperparameter settings, **highlighting overfitting** in the local models.
>
> We believe this revision addresses the concern and provides a clearer understanding of how personalization overfitting manifests. Thank you for the valuable feedback.

---

> > ### Comment · Reviewer_sf8C · 2024-11-27
> >
> > Thanks for the authors' response! Some of my concerns have been addressed, and I have adjusted my rating accordingly. Good luck!

---

> > > ### Author Response · Authors · 2024-11-27
> > >
> > > Thank you for your timely feedback and adjusted rating. We sincerely appreciate your constructive comments, which have helped clarify the contribution and novelty of our work. We will further refine the paper in the final revision. Thank you!

---

### Official Review · Reviewer_pW35 · 2024-11-04

**Soundness:** 2
**Presentation:** 3
**Contribution:** 2
**Rating:** 5
**Confidence:** 5

**Summary:**

In this work, the authors study the trade-off between personalization and global generalization in Federated Learning (FT).  They found that personalized finetuning (PFT)  causes overfitting and inconsistent performance across distribution shifts in federated learning. Furthermore, the authors identify LP-FT, which is a two-stage fine-tuning method that first performs linear probing and then full fine-tuning to improve local adaptation while preserving global features. The evaluation proves that LP-FT consistently adapts distribution shifts and outperforms existing work.

**Strengths:**

1. It is interesting and meaningful to study federated learning with distribution shifts, and the authors provide a practical method to enhance the OOD issues.
2. The adaption of LP-FT is general for diverse datasets and distribution shifts, making it versatile for real-world FL scenarios.

**Weaknesses:**

A substantive assessment of the weaknesses of the paper. Focus on constructive and actionable insights on how the work could improve towards its stated goals. Be specific, avoid generic remarks. For example, if you believe the contribution lacks novelty, provide references
and an explanation as evidence; if you believe experiments are insufficient, explain why and exactly what is missing, etc.

1. The related work is insufficient in two categories. Firstly, some state of the art  Parameter-Efficient Fine-Tuning methods, e.g., LoRA[1], IA3[2], are not discussed and compared. Secondly, some of the FL methods considering distribution shifts are not considered, e.g., FedTHE[3] and FedIIR[4].

2. The contribution and novelty need to be strengthened, e.g., except for making LP-FT adapts to decentralized datasets in FL scenarios, what else is vital and seldom studied. At present, the contribution of LP-FT is solid in centralized learning but not customized for FL.

3. The computation cost of introducing LP-FT is overwhelming, please provide both empirical and theoretical analysis to quantify it.

4. The empirical studies could be more extensive. On the one hand, the authors could illustrate some experimental details like which cifar10-C is evaluated. On the other hand, the authors are expected to study non-IID scenarios in FL, because in FL, modeling IID decentralized data is similar with centralized training.

5. The experiment analysis is not conclusive, e.g., in Table 1, why LP-FT outperforms other methods on the condition that some results are worse than baselines.

6. The theoretical analysis overlooks the data is heterogeneous.

[1] Liu H, Tam D, Muqeeth M, et al. Few-shot parameter-efficient fine-tuning is better and cheaper than in-context learning[J]. Advances in Neural Information Processing Systems, 2022, 35: 1950-1965.
[2] Hu E J, Shen Y, Wallis P, et al. Lora: Low-rank adaptation of large language models[J]. arXiv preprint arXiv:2106.09685, 2021.
[3] Jiang L, Lin T. Test-Time Robust Personalization for Federated Learning[C]//The Eleventh International Conference on Learning Representations.
[4] Guo Y, Guo K, Cao X, et al. Out-of-distribution generalization of federated learning via implicit invariant relationships[C]//International Conference on Machine Learning. PMLR, 2023: 11905-11933.

**Questions:**

1. How LP-FT outperform existing Parameter-Efficient Fine-Tuning methods for federated scenarios?
2. Can authors discuss the most promising contributions for this work?
3. Whether this overfitting and inconsistent issues happen for language data?

---

> ### Author Response · Authors · 2024-11-25
>
> **Response to W1.1 & Q1: Clarification on our focused PFT vs. reviewer-suggested PEFT**
>
> Thank you for your feedback. We would like to clarify that our paper primarily focuses on the **personalized fine-tuning (PFT)** rather than the **parameter-efficient fine-tuning (PEFT)** on FL.
>
> PFT focuses on balancing the global and local risk during the **final personalized fine-tuning** stage of the FL (note that the fine-tuning here indicates different clients continue to train the global model on their individual local data). Such fine-tuning can be PEFT and non-PEFT. Given that our work is not centered on efficient fine-tuning foundation models, our primary focus is on effectively managing the trade-off between local and global risks.
>
> Although our primary focus differs, we address the reviewer’s request by showing results for PEFT. We adopted two widely recognized PEFT methods: **LoRA** [1] an **Adapter** [2]. Both methods were fine-tuned with meticulous adjustments to their learning rates on the ViT, as detailed in Table below. It is worth noting that the PEFT methods are not specifically designed for PFT. As expected, their performance significantly underperforms our proposed LP-FL.
>
>
>
> | **PEFT Method**       | Local  | Global | Avg.   |
> |------------------------|--------|--------|--------|
> | LoRA (lr=1e-3)        | 41.54  | 26.75  | 26.87  |
> | LoRA (lr=1e-4)        | 42.01  | 26.41  | 27.18  |
> | Adapter (lr=1e-3)     | 48.35  | 39.28  | 38.08  |
> | Adapter (lr=1e-4)     | 49.07  | 39.83  | 38.36  |
> | LPFT                  |  **68.5** |  **57.52** |  **53.52** |
>
> **Table:** Different PEFT compared on DomainNet with ViT.
>
> Reference
> - [1] Hu, Edward J., et al. "Lora: Low-rank adaptation of large language models." arXiv preprint arXiv:2106.09685 (2021).
> - [2] Houlsby, Neil, et al. "Parameter-efficient transfer learning for NLP." International conference on machine learning. PMLR, 2019.
>
>
> ---
>
> **Response to W1.2: Clarification on our focused PFT vs. other FL methods that consider distribution shift.**
>
> **1. Clarification on our PFT’s focus and setting**
> Although our method also considers the distribution shift problem, our focus is fundamentally different from the reviewer's pointed FL methods. Unlike most existing FL methods on distribution shift that address the issue through the whole FL training process, our approach focuses on the **critical yet underexplored final local training stage** after the global model is obtained. This makes our method a **versatile solution**, **independent of the global model training process**. This shift not only introduces a **fresh perspective on FL personalization** but also **demonstrates significant theoretical and empirical insights**.
>
> Specifically, FedIIR is a GFL method that modifies the entire FL training pipeline across all stages, whereas our PFT setting focuses exclusively on the final personalized fine-tuning stage, operating in parallel to GFL methods. FedTHE addresses a completely different scenario, focusing on test-time adaptation to improve personalization. Our PFT setting, however, emphasizes balancing global and local performance during the final personalized fine-tuning stage.
>
> **2. Discussion on related work **
> Although the focus and settings differ, we have not omitted related works on FL for distribution shift, as evidenced by our extensive discussion on heterogeneous FL in the first paragraph of Section 2 (Related Work).

---

> ### Author Response · Authors · 2024-11-25
>
> **Response to W2 & Q2: Highlighting Contribution and Novelty**
>
> We would like to clarify that the novelty and contribution of our work extend beyond simply adapting an existing method (LP-FT [1]) to a new scenario (PFT). Our key contributions lie in the **methodological and theoretical novelty**, as well as **empirical findings, benchmarking, and analysis**. These efforts provide **deeper insights** and **fine-grained evaluations** of diverse distribution shifts, demonstrating the effectiveness of LP-FT in the **new PFT setting**.
>
> **Methodological Novelties (See Section 1 Contribution Insight)**
>
> **1. Simple yet Non-Trivial Method: Novel Focus on Final-Stage LP-FT in FL**
>
> While our method may appear straightforward, its novelty lies in **strategically shifting the insertion phase** of LP-FT to the **final local training stage** in FL. Unlike most existing PFL methods that require modifications to the entire training process, including both General FL (GFL) training and local personalization, our approach focuses on the **critical yet underexplored final local training stage**. This makes our method a **versatile solution**, **independent of the GFL training process**. This shift not only introduces a **fresh perspective on FL personalization** but also **demonstrates significant theoretical and empirical insights**.
>
> We also respectfully wish reviewers to agree that **Simplicity does not diminish the novelty of our method**; instead, it enhances its practical impact and potential for widespread applicability in real-world FL scenarios. A simple yet effective method, like ours, is a strength that enables broader adoption while providing substantial improvements over state-of-the-art techniques.
>
> **2. Pioneering LP-FT to FL:**
>
> To the best of our knowledge, this work is **the first to systematically explore and leverage** the potential of the simple yet effective **LP-FT** method within the **FL** framework. This represents a significant contribution, as it opens up a new line of research exploring the application and adaptation of LP-FT in addressing the unique challenges of FL, such as data heterogeneity and distribution shifts.
>
> **Theoretical Novelties (See Section 1 Theory Contribution)**,
>
> **1. Our theoretical setting and assumption differ significantly from that of [1].**
> While [1] assumes a **centralized setting**, we focus on **personalized fine-tuning (PFT)** in FL, a fundamentally different scenario. In our case, the pre-trained model is trained on data from all clients, reflecting a global client distribution. This contrasts with the centralized setting in [1], where the pre-trained model is trained without any notion of a global client distribution.
>
> **2. Our theoretical analysis diverges significantly from that of [1].**
> In [1], the assumption is that a single ground-truth data-generating function underlies all data points. This assumption makes it challenging to study LP-FT in different distribution shift settings in PFT for FL (e.g., concept shift). In contrast, we assume that each client has a distinct ground-truth function, enabling the exploration of personalized fine-tuning in FL under varying distribution shifts. Specifically, the way we define the data-generating function (**Assumption 4.1**), with unique linear heads before pre-training $V_i^* = \begin{bmatrix} {V_{com}^*}^T & \lambda e_i^T \end{bmatrix}^T$ for each client and the way we define data heterogeneity $x_i = e_i + \epsilon n, \quad \text{with} \quad n \sim \mathcal{N}(0, I).$ allow us to study both concept shift and covariate shift using gradient descent to compare FT and LP-FT in terms of global performance. This approach provides new insights into the challenges of PFT in FL. In other words, the main challenge we address in our theoretical analysis lies in incorporating assumptions that align the theoretical framework with our specific setup. Our analysis connects the superior global performance of the LP-FT method to the concept of feature distortion. This novel perspective offers valuable insights into the effectiveness of LP-FT in FL through the lens of feature distortion.
>
> **3. In-depth Integrating Theoretical Analysis and Empirical Studies:**
> We have conducted additional empirical studies to substantiate the theoretical analysis on the heterogeneity level $\lambda$. These include a comprehensive investigation with diverse ablation studies across varying levels of label-flipping ratios (**Section 5**). This **significantly extends the scope and granularity of distribution shifts** explored in the original LP-FT paper [1], providing deeper insights into its robustness and applicability under heterogeneous conditions.
>
> **[Continued below]**

---

> ### Author Response · Authors · 2024-11-25
>
> **[Continued from above]**
>
> **Response to W2 & Q2: Hilighting Contribution and Novelty**
>
> **Empirical Contributions (See Section 1 Contribution Evaluation):**,
>
> **1. New Observations: Detailed Analysis on Overfitting in PFT:**
> As shown in **Figure 2**, we analyze commonly used fine-tuning strategies and demonstrate their **tendency to overfit**, even with carefully tuned hyperparameters, across three different types of distribution shifts. These findings highlight a critical gap in prior FL research, where such tendencies **have not been systematically explored**.
>
> **2. Comprehensive Benchmarking PFT:**
> Our experiments span **seven diverse datasets** (CIFAR-10-C, CIFAR-10, Digit5, DomainNet, Chexpert, CelebA, CIFAR-10), addressing **four types of distribution shifts** (input-level shift, feature shift, spurious correlation, and label shift, as detailed in **Appendix Table 6**). We evaluate **five Personalized Fine-Tuning (PFT) methods** using **five robust metrics**, providing a thorough and comprehensive analysis.
>
> **3. Establishing New Ablation Studies to Support Theoretical Results:**
> In **Figure 4**, we conduct detailed analysis on feature distortion relation to overfitting and in **Table 2**, we conduct large ranges of label flipping to support the theoretical analysis validating our theoretical insights in low heterogeneity regimes and further demonstrating LP-FT’s superior performance in scenarios with higher heterogeneity.
>
>
>
> Reference
> - [1] Ananya Kumar, Aditi Raghunathan, Robbie Matthew Jones, Tengyu Ma, and Percy Liang. Fine-tuning can distort pretrained features and underperform out-of-distribution. In ICLR. OpenReview.net, 2022.

---

> ### Author Response · Authors · 2024-11-25
>
> **Response to W3: Clarification on Computational Cost Concerns for LP-FT**
>
> We **respectively disagree that LP-FT’s computation cost is overwhelming**.
> We start by clarifying the PFT training pipeline. First, PFT has nothing to do with the GFL, thus LP-FT has the same computational cost compared to FedAvg for GFL. Second, our LP-FT is only **integrated exclusively into the PFT stage**.
>
> **Theoretically**, compared to full fine-tuning (FT), LP-FT introduces an **additional operation of linear probing (LP)** with a computational complexity of $O(nmd)$ when applying gradient descent on a dataset of size $n$, using a linear head with input dimension $d$ and output dimension $m$. If the LP step is performed for $e$ epochs, the computational complexity becomes $O(e \cdot nmd)$. We have incorporated this theoretical analysis of the computational complexity of LP-FT compared to FT in the appendix of the revised paper.
>
> **Empirical Evaluation**
> Empirically, we evaluate the **FLOPs** for one step (forward and backward for one batch) on Digit5 dataset. Specifically, we compare LP for 1 step with batch size of 128 and FT for 1 step with a batch size of 128. Since the computational cost of LP is only in the order of $O(10^{-3})$ of FT.  LP is inherently more efficient in its operations. We measured the computational cost using FLOPs for one step (batch size of 128) on ResNet18 with the Digit5 dataset. Our analysis shows that FT requires $6.444 \times 10^{12}$ compared to $6.443 \times 10^{12}$ for LP.

---

> ### Author Response · Authors · 2024-11-25
> **Official Comment by Authors**
>
> **Response to W4:Clarification on comprehensive experiments & Addressing misunderstanding on experimental settings**
>
> **Clarification on the comprehensiveness of experiments:**
>
> We greatly appreciate the reviewer's feedback and would like to respectfully clarify that we believe **our empirical studies are comprehensive**, a perspective that has also been recognized by other reviewers. Our experiments cover **seven diverse datasets** (CIFAR-10-C, CIFAR-10, Digit5, DomainNet, Chexpert, CelebA, CIFAR10), **four types of distribution shifts** (input-level shift, feature shift, spurious correlation, label shift, as the label shift is detailed in **Appendix Table 6**), **five Personalized Fine-Tuning (PFT) methods**, and employ **five evaluation metrics** to provide a robust analysis. This comprehensiveness has been recognized by multiple reviewers:
>
> - Reviewer **7VAu** remarked that
>     > “Extensive experiments and theories were conducted.”
>
> - Reviewer **gFiS** highlighted the
>     > “comprehensive experimental observation and theoretical analysis.”
>
> - Reviewer **sf8C** noted the evaluation covers
>     > “multiple data heterogeneity scenarios.”
>
> We believe these assessments strongly support the thoroughness of our empirical evaluations.
>
> **Regarding CIFAR-10-C:**
>
> CIFAR-10-C is a **well-established benchmark** in the research community for evaluating distribution shifts and federated learning settings. It comprises corruption patterns applied to CIFAR-10 to simulate real-world robustness challenges, making it a natural choice for our work. While it is **unclear what is meant by “which CIFAR-10-C,”** we have included detailed descriptions and references to CIFAR-10-C in **Appendix Table 3** to assist readers who may be less familiar with this benchmark.
>
>
> **Clarification and Correction of Fundamental Factual Misunderstanding:**
>
> We **respectfully disagree** with the comments and clarify that we **HAVE thoroughly addressed non-IID scenarios in FL**. We emphasize that data heterogeneity and distribution shifts, inherent to non-IID settings, are central themes of our work. This is **clearly highlighted throughout our paper**:
>
> - Abstract: Lines 16–19
> - Section 1: Lines 75–79
> - Section 1: Lines 97–99
> - Section 1: Lines 103–106
> - Section 2: Lines 110–125
> - Section 3.1: Lines 162–176
> - Section 3.2: Lines 181–192
> - Section 3.3: Lines 214–235
> - Section 3.4: Lines 315–317
> - Section 4: Assumption 4.1 (Line 416)
> - Section 4: Lines 477–492
> - Section 4: Lines 500–521
> - Section 6: Line 539
>
> **In summary**. Our experiments encompass diverse Non-IID settings, including various distribution shift types and degrees of data heterogeneity. Throughout the original paper, we have clearly conveyed these Non-IID settings.

---

> ### Author Response · Authors · 2024-11-26
>
> **Response to W5: Results Explanation and Takeaway**
>
> We respectfully disagree that our experiments are not conclusive. We would like to start by clarifying our evaluation criteria, and then highlight the takeaways from our results.
>
>
> **Clarification of Evaluation Criteria:**
>
> LP-FT's superiority over baseline methods is primarily evaluated based on Global Accuracy, the Worst Accuracy, and Avg. Accuracy, as these align with our focus on feature distortion effects. We emphasize that LP-FT consistently outperforms the baselines across most scenarios under these criteria.
>
> **Highlighting Key Improvements on LP-FTs**
>
> To provide more concrete evidence of LP-FT's efficacy, we point out the following improvements:
> In CIFAR10-C (input shift), LP-FT improves upon the best baseline (Proximal FT) by 1.77% in Global Accuracy, 0.51% in Worst Accuracy, and 1.36% in Average over local and global accuracy.
> In DomainNet (feature shift), LP-FT achieves an improvement of 1.47% in Global Accuracy, 1.33% in Worst Accuracy, and 0.92% in Average over local and global accuracy over the strongest baseline.
> Across CelebA (spurious correlation), LP-FT demonstrates 2.21% gains in Global Accuracy, 4.39% in Worst Accuracy, and 2.04% in Average over local and global accuracy.
>
> **Justifications for Limited Cases Where LP-FT Did Not Outperform:**
>
> For the cases where LP-FT does not surpass the baselines, we have provided detailed explanations in Section 3.4 results analysis of the paper. LP-FT already surpasses 90% of the baselines. LP-FT's design prioritizes preventing feature distortion, which is deeply discussed in the theoretical analysis. This aligns with the overarching objectives of this study. While certain trade-offs (e.g., performance in niche scenarios) are observed, they are an intentional choice to achieve better overall robustness and adaptability across diverse datasets and clients.
>
> ---
>
> **Response to W6: Our Theoretical Analysis Exactly Addresses Data Heterogeneity**
>
> There appears to be a misunderstanding that the reviewer thought ``.theoretical analysis overlooks the data is heterogeneous”. We would like to clarify that our theoretical analysis IS CENTERED ON  data heterogeneity. As stated in **Line 466-469** of our original submission (**Line 478–481** of our revised submission), we introduce covariate shift with the explicit assumption that data is heterogeneous.
>
> Our theoretical analysis incorporates data heterogeneity based on the specific setting under consideration, whether it involves concept shift or a combination of concept and covariate shifts. In **Section 4.1**, we study the global performance of LP-FT under concept shift alone, assuming data is not heterogeneous (i.e., $\mathbb{E}_{x \sim \mathcal{D}_i}[x x^T] = I_d$ for all clients $i \in [C]$). In **Section 4.2**, however, we extend our analysis to encompass both concept and covariate shifts, incorporating data heterogeneity by allowing each client's data to be generated as $x_i = e_i + \epsilon n$, where $n \sim \mathcal{N}(0, I)$ and $e_i$ represents a client-specific shift. Therefore, our theoretical analysis considers or excludes data heterogeneity in accordance with each section’s focus.
>
> ---
>
> **Response to Q3**
>
> Thanks for the suggestions on exploring text data. In this work, we follow the common stream of heterogeneous FL studies and closely related work on vision tasks. It would be an interesting future extension work.
>
> Our focus on vision tasks is **consistent with the original LP-FT paper** [1] and **its extended analysis** [2], both of which are centered on vision data. By following this established scope, we aim to build upon and extend the insights of prior research within the same modality.
>
> Reference:
> - [1] Kumar, A., Raghunathan, A., Jones, R., Ma, T., & Liang, P. (2022). Fine-tuning can distort pretrained features and underperform out-of-distribution. arXiv preprint arXiv:2202.10054.
> - [2] Trivedi, P., Koutra, D., & Thiagarajan, J. J. (2023). A closer look at model adaptation using feature distortion and simplicity bias. arXiv preprint arXiv:2303.13500.

---

> > ### Comment · Reviewer_pW35 · 2024-11-27
> > **Reply to the authors**
> >
> > Your feedback helped clarify some misunderstandings, I would like to raise my score to 5. But most critical issues remain unresolved.
> > First, the authors seem unfamiliar with federated learning on heterogeneous data [1]. Kindly point out the specific data heterogeneity simulation you are using. **In the current version**, there are no clear demonstrations of this in your main experiments. Although you reference multiple aspects of heterogeneity in your response, most of these discussions focus on global and local performance. However, the core issue of data heterogeneity lies in the non-IID nature among clients, which is **only validated in D.3**.
> > In this context, if all clients share identical data distributions, your work essentially reduces to the original effectiveness of LP-FT in centralized modeling. This is because the **best-pretrained model you assume** is merely transferred to each client individually, without any meaningful collaboration between clients. **Collaboration among clients**, however, is the essence of federated learning [2].
> > Secondly, your experiments and theoretical analysis rely on a strong assumption: the existence of a **perfect pretrained model**. For instance, (1) in the proof of Lemma 4.3, you discuss each client in isolation, and (2) in the proof of Theorem 4.4, it is assumed that the pretrained model perfectly captures the feature extractor matrix B∗B^*. However, the impact of data heterogeneity significantly complicates the realization of such perfect pretrained models. Currently, your work provides no contributions toward addressing this limitation.
> > In summary, while the authors may consider their work to be meaningful, it does not demonstrate a direct contribution to the field of federated learning.
> >
> > [1] Li T, Sahu A K, Zaheer M, et al. Federated optimization in heterogeneous networks[J]. Proceedings of Machine learning and systems, 2020, 2: 429-450.
> > [2]McMahan B, Moore E, Ramage D, et al. Communication-efficient learning of deep networks from decentralized data[C]//Artificial intelligence and statistics. PMLR, 2017: 1273-1282.

---

> > > ### Author Response · Authors · 2024-11-30
> > >
> > > We appreciate your detailed feedback and the opportunity to address the key aspects of our work, particularly regarding the non-IID setup in FL in experiments and theory. Below, we clarify these points from both **experimental (Part A)** and **theoretical perspectives (Part B)**.
> > >
> > > ---
> > >
> > > **A Experiment Clarifications**
> > >
> > > In this section, we focus on addressing misunderstandings regarding our non-IID setting in experiments.
> > >
> > > ---
> > >
> > > **A.1 Data Heterogeneity in Experiments**
> > >
> > > > “First, the authors seem unfamiliar with federated learning on heterogeneous data [1]. Kindly point out the specific data heterogeneity simulation you are using.”
> > >
> > > We are, of course, familiar with [1] and its data heterogeneity setup. As clearly stated in Section 3.1, our work mainly focuses on **covariate shift and concept shift**, which represent important types of data heterogeneity distinct from the label shift examined in [1]’s real vision dataset simulation. We respectfully urge the reviewer to acknowledge this line of **more recent heterogeneous FL work** **[2, 3, 4]** and recognize that our exploration of data heterogeneity aligns with these broader perspectives.
> > >
> > > Our setup (**Section 3.1 Problem Definition, Section 3.4 Experimental Setup, Appendix D.3)** explicitly includes description on our data heterogeneity setting, including covariate shift, concept shift, and label shift. Covariate and concept shifts are formalized in **Section 3.1**, with datasets and split strategies detailed in **Section 3.2** and **Appendices C.2 and C.3**, demonstrating the non-IID nature of client data. These shifts generate realistic heterogeneous scenarios commonly studied in FL **[2, 3, 4]**.
> > >
> > > ---
> > >
> > > **A.2 Global and Local Performance Metrics**
> > >
> > > > “Although you reference multiple aspects of heterogeneity in your response, most of these discussions focus on global and local performance.”
> > >
> > > We respectfully clarify that the evaluation of **global and local performance are important performance metrics** of **personalization** in FL and are directly tied to the underlying data heterogeneity [5, 6, 7, 8].
> > >
> > > The evaluation of global and local performance is central to personalization in Federated Learning (FL). During General Federated Learning (**GFL**), global performance is prioritized, often at the cost of local performance. Conversely, in Personalized Federated Learning (**PFL**), local performance is emphasized, often leading to trade-offs with global performance.
> > >
> > > **These metrics inherently reflect data heterogeneity**. If clients shared IID data, global and local performance would theoretically align and would be nearly identical empirically. Thus, distinguishing between these metrics is essential for evaluating the trade-offs caused by non-IID client data distributions in FL. **Section 3.1 (Lines 159–165)** explains the importance of these metrics for assessing the trade-offs between personalization (local performance) and generalization (global performance) in non-IID FL settings.
> > >
> > > ---
> > >
> > > **A.3 Non-IID Nature Beyond Label Shift**
> > >
> > > > “The core issue of data heterogeneity lies in the non-IID nature among clients, which is only validated in D.3.”
> > >
> > > We want to kindly clarify that **data heterogeneity in FL is not limited to label shift**. Covariate and concept shifts modify input data distribution P(X) and the conditional distribution of label Y given input data X, P(Y | X), respectively (**Section 3.1 Lines 169–182**). These shifts create non-IID data across clients, as evidenced by our dataset splits and evaluation strategies. These non-IID setups are widely recognized in FL personalization research **[2, 3, 4]**.
> > >
> > > ---
> > >
> > > **A.4 Misrepresentation of Experimental Setup**
> > >
> > > > “If all clients share identical data distributions, your work essentially reduces to the original effectiveness of LP-FT in centralized modeling.”
> > >
> > > This statement is factually incorrect. All clients in our setup have distinct, non-IID data distributions. Our evaluation explicitly models client-level heterogeneity, as detailed in **Sections 3.1 and 3.4**, making it fundamentally different from centralized modeling.

---

> > > > ### Author Response · Authors · 2024-11-30
> > > >
> > > > **A.5 No Assumption on Best Pretrained Model**
> > > >
> > > > > “The best-pretrained model you assume is merely transferred to each client individually, without any meaningful collaboration between clients.”
> > > >
> > > > We clarify that we **NEVER assume the perfect pretrained model in our experiment**. We also respectfully disagree that our problem is just about transferring the best-pre trained model to each client individually.
> > > >
> > > > **First**, we hope the reviewer acknowledges that **our work is NOT merely about transferring the best-pretrained model to each client**. Instead, it addresses the challenging and underexplored question in FL of how to ‘better fine-tune the well-trained global model obtained through client collaboration without sacrificing too much global performance,’ especially under the data heterogeneity present across clients. This is coupled with the consideration of both local and global performance metrics, which are essential in evaluating personalization in FL.
> > > >
> > > > **Unlike centralized LP-FT**, which focuses on fine-tuning a pretrained model on a single domain and evaluating unseen domain generalization, our work explores unique and novel questions in FL: ‘**Can we perform LP-FT in personalized fine-tuning with a GFL model?**’ and ‘**Can the resulting model achieve better global and local performance?**'
> > > >
> > > > **Second**, the meaning collaboration features in our study are bounded with:
> > > > - The **initialization** of LP-FT is a **GFL-pretrained mode from client collaboration**. Such initialization is also critical in our theoretical analysis.
> > > > - The **evaluation metrics** on the LP-FT models consider the generalization across different clients (**global performance**)
> > > >
> > > > **Specifically**, our pipeline includes a general FL (GFL) stage to collaboratively train a shared global model, and a personalized fine-tuning (PFT) stage to locally train a local model for personalization without further communication rounds. The GFL stage includes client collaboration. a shared global model is trained using the FedAvg algorithm (a general FL method). During the personalized fine-tuning stage (PFT), clients adapt the global model to their local data while balancing global and local performance to prevent overfitting.
> > > >
> > > > This process is detailed in **Figure 1(a), Section 1 (Lines 70–73), Section 3.3 (Lines 215–237), and Section 3.4 (Lines 264–315)**. The collaborative aspect is inherent in the shared global model and subsequent evaluation of global and local performance.
> > > >
> > > > ---
> > > >
> > > > **In Summary**, we hope these clarifications address the reviewer’s concerns and clarify the **non-IID setup** and PFT pipeline in our experiments.
> > > >
> > > > ---
> > > >
> > > > Experiment Part References
> > > > - [1] Li, Tian, et al. "Federated optimization in heterogeneous networks." Proceedings of Machine learning and systems 2 (2020): 429-450.
> > > > - [2] Li, Xiaoxiao, et al. "Fedbn: Federated learning on non-iid features via local batch normalization." arXiv preprint arXiv:2102.07623 (2021).
> > > > - [3] Wu, Shanshan, et al. "Motley: Benchmarking heterogeneity and personalization in federated learning." arXiv preprint arXiv:2206.09262 (2022).
> > > > - [4] Ogier du Terrail, Jean, et al. "Flamby: Datasets and benchmarks for cross-silo federated learning in realistic healthcare settings." Advances in Neural Information Processing Systems 35 (2022): 5315-5334.
> > > > - [5] Liang, Paul Pu, et al. "Think locally, act globally: Federated learning with local and global representations." arXiv preprint arXiv:2001.01523 (2020).
> > > > - [6] Chen, Hong-You, and Wei-Lun Chao. "On bridging generic and personalized federated learning for image classification." arXiv preprint arXiv:2107.00778 (2021).
> > > > - [7] Chen, Minghui, et al. "FedSoup: improving generalization and personalization in federated learning via selective model interpolation." International Conference on Medical Image Computing and Computer-Assisted Intervention. Cham: Springer Nature Switzerland, 2023.
> > > > - [8] Deng, Wenlong, Christos Thrampoulidis, and Xiaoxiao Li. "Unlocking the potential of prompt-tuning in bridging generalized and personalized federated learning." Proceedings of the IEEE/CVF Conference on Computer Vision and Pattern Recognition. 2024.

---

> > > > > ### Author Response · Authors · 2024-11-30
> > > > >
> > > > > **B. Theory Clarifications**
> > > > >
> > > > > We would like to clarify the data heterogeneity addressed in our theoretical framework, along with the rationale and justification for our theoretical assumptions, as outlined below.
> > > > >
> > > > > ---
> > > > >
> > > > > **B.1 Restatement on our data heterogeneity consideration in theory**
> > > > >
> > > > > > ”However, the core issue of data heterogeneity lies in the non-IID nature among clients, which is only validated in D.3. In this context, if all clients share identical data distributions, your work essentially reduces to the original effectiveness of LP-FT in centralized modeling.”
> > > > >
> > > > > **B.1.1 Clarification on data heterogeneity in our theory**:
> > > > >
> > > > > We respectfully clarify that our **theoretical analysis explicitly incorporates non-IID data distributions among clients**, addressing the data heterogeneity inherent in federated learning.
> > > > >
> > > > > **Covariate and Concept Shift in our theory**: We emphasize that our theoretical analysis does not universally assume that clients share identical data distributions. Instead, our analysis is structured into two distinct scenarios: one considering concept shift and the other addressing combined covariate-concept shift. Each scenario is analyzed under its own specific assumptions, where the assumption of identical data distributions may or may not hold, depending on the context.
> > > > >
> > > > > In **Section 4.2**, we introduce covariate shift under the assumption that clients have different data distributions, specifically $x_i = e_i + \epsilon n$ (see **Lines 466-469** in our original submission and **Lines 478-481** in our revised submission). This reflects the non-IID nature of clients’ data distributions. Consequently, our work does not reduce the effectiveness of LP-FT in centralized modeling, as suggested by the reviewer.
> > > > >
> > > > > **B.1.2 Clarification on the federated setting in our theory**:
> > > > >
> > > > > Our work addresses a **fundamentally different scenario from the centralized setting**: PFT in the FL context. Here, the pre-trained model is trained on data from all clients, reflecting a global client distribution. After fine-tuning the model on each client, we evaluate its global performance using data from all clients. Notably, in **Theorems 4.4 and 4.5**, we compare the global loss of LP-FT and FT by computing the loss across all clients, each with distinct data-generating functions. This distinction—the presence of multiple clients with varying data-generating functions—sets our work apart from the centralized setting.
> > > > >
> > > > > In the **centralized setting described in [1]**, the **pre-trained model is trained without consideration of a global client distribution**. Instead, it is assumed that a single ground-truth data-generating function underlies all data points, as there are no distinct clients. **By contrast**, our work assumes that each client has a unique ground-truth data-generating function, allowing us to study PFT in FL under diverse distribution shifts.
> > > > >
> > > > > Additionally, in **Section 4.2**, we explicitly model covariate shift by assuming distinct data distributions $x_i = e_i + \epsilon n$ across clients. This highlights the inherent data heterogeneity in FL scenarios.
> > > > > Consequently, our work cannot be reduced to centralized modeling. Instead, it explicitly accounts for non-IID distributions and diverse data-generating functions across clients, a key component of our theoretical framework.

---

> > > > > > ### Author Response · Authors · 2024-11-30
> > > > > >
> > > > > > **B.2 Rationale of our theoretical assumptions**
> > > > > >
> > > > > > **B.2.1 Clarification on the “prefect pretrained model”**
> > > > > >
> > > > > > > “Secondly, your experiments and theoretical analysis rely on a strong assumption: the existence of a perfect pretrained model.”
> > > > > >
> > > > > > **We DO NOT assume the existence of a perfect pretrained model.** Instead, our theoretical framework assumes the recovery of the true feature extractor (B_0 = B_*) during GFL training, **consistent with prior theoretical work [1]**. **The recovery of true features does not imply a perfect pretrained model.**
> > > > > >
> > > > > > To clarify, first our experimental setting does not rely on the assumption of a perfect (pre-trained) model. In fact, the pre-trained model used in our work is derived from the GFL stage, where all clients collaboratively train a shared model. This collaboratively trained model then serves as the pre-trained model for the subsequent fine-tuning phase conducted by each client. **The well-trained global model from the GFL stage is not guaranteed to perform optimally on each client’s local data**, which underscores the need for personalized fine-tuning. This is evident in **Figure 2**, where global accuracy consistently decreases during the PFT stage as local models become more tailored to individual client data. Consequently, the notion of a perfectly pre-trained model is unrealistic in either local or global scenarios. Therefore, our experimental setup does not rely on any assumption of a perfect pre-trained model.
> > > > > >
> > > > > > **B.2.2 Clarification on the pipeline of collaborative training and personalized fine-tuning in theory**
> > > > > >
> > > > > > > “However, the impact of data heterogeneity significantly complicates the realization of such perfect pretrained models.”
> > > > > >
> > > > > > **Our experiments do not rely on the assumption of a perfect pretrained model.** A more accurate understanding of the relationship between theory and experiments is that our theoretical analysis of LP-FT provides **justification** for its empirical effectiveness in the PFT setting. Furthermore, in theory, this global model is not guaranteed to perform optimally on each client’s local data, necessitating personalized fine-tuning.
> > > > > >
> > > > > > **Specifically**, regarding the theoretical analysis, we introduced two key assumptions: (1) a two-layer data-generating function (**Assumption 4.1**) and (2) a two-layer model structure (**Assumption 4.2**). These assumptions allow us to study the effects of pre-trained feature representations and linear heads. To analyze our setting (**PFT following the GFL stage**), we assume that the pre-trained feature extractor recovers the true feature extractor, i.e., $B_0 = B_*$. However, the pre-trained linear head only captures the common part of the linear head, i.e., $V_0 = [V_{\text{com}}^*, 0]^T$. This assumption of recovering the feature extractor is adopted to explain the impact of feature distortion introduced during the PFT stage and its effect on global performance.
> > > > > >
> > > > > > Notably, **the assumption of recovering the true feature extractor ($B_0 = B_*$) is consistent with Proposition 3.7 in [1]**, where it is used to compare the performance of LP-FT and FT after fine-tuning. The insights provided in our paper regarding feature distortion after fine-tuning in the FL setting are rooted in a theoretical framework that models clients’ unique data-generating functions with a shared feature extractor. These assumptions, including the coexistence of shared features and heterogeneity across clients’ data-generating functions, enable us to capture the unique characteristics of our PFT setting.
> > > > > >
> > > > > > **B.2.3 Misrepresentation on our assumption**
> > > > > >
> > > > > > > “Secondly, your experiments and theoretical analysis rely on a strong assumption: the existence of a perfect pretrained model.”
> > > > > >
> > > > > > To **reiterate**, we **DO NOT** assume a perfect pretrained model but rather a perfectly recovered feature extractor. Additionally, our experiments are **NOT** based on this strong assumption. Our theoretical analysis of LP-FT serves to justify its empirical effectiveness within the PFT setting.
> > > > > >
> > > > > > We want to **clarify again** that the assumption of recovering the true feature extractor ($B_0 = B_*$) is distinct from the existence of a perfect pre-trained model (mentioned by the reviewer above). The pre-trained model consists of both a feature extractor $B$ and a linear head $V$. Thus, our work does not rely on the assumption of a perfect pre-trained model.
> > > > > >
> > > > > > ---
> > > > > >
> > > > > > **In summary**, we clarify the consideration of data heterogeneity in our theoretical framework and provide a detailed rationale and justification for our assumptions.

---

> > > > > > > ### Author Response · Authors · 2024-11-30
> > > > > > >
> > > > > > > ---
> > > > > > >
> > > > > > > Theory Part Reference:
> > > > > > > - [1] Kumar, A., Raghunathan, A., Jones, R., Ma, T., & Liang, P. (2022). Fine-tuning can distort pretrained features and underperform out-of-distribution. arXiv preprint arXiv:2202.10054.
> > > > > > >
> > > > > > > ---
> > > > > > >
> > > > > > > **Conclusion**
> > > > > > > We hope these clarifications on both experiments and theory address your concerns. Our paper explicitly handles a broad range of **non-IID settings**, covering both label, covariate, and concept shifts, and demonstrates the efficacy of LP-FT in addressing these challenges. Our **theoretical assumption is also reasonable** which aligns with previous theoretical paper’s assumptions.
> > > > > > >
> > > > > > > If any uncertainties remain despite our detailed clarification, including references to original text statements, descriptions, and related work, we kindly encourage the reviewer to consider **the acknowledgments from other reviewers regarding our non-IID FL setting**.
> > > > > > >
> > > > > > > > Reviewer 7VAu: ‘The authors perform comprehensive evaluations on various scenarios (e.g., different types of data heterogeneity, datasets, and models).’
> > > > > > >
> > > > > > > > Reviewer gFiS: ’[The work focuses] on feature distribution shift scenarios.’
> > > > > > >
> > > > > > > > Reviewer sf8C: ’[The paper addresses] a long-standing and challenging issue in FL, i.e., data heterogeneity.’
> > > > > > >
> > > > > > > and **their recognition of our theoretical contributions**:
> > > > > > >
> > > > > > > > Reviewer 7VAu: ‘Theories were conducted to verify the advantages of the proposed method in preserving the generality when personalizing models.’
> > > > > > >
> > > > > > > > Reviewer gFiS: ’[This work] provided a solid theoretical analysis using a two-layer linear network.’
> > > > > > >
> > > > > > > > Reviewer sf8C: ‘The paper conducts both experimental evaluation and theoretical analysis to demonstrate the method’s effectiveness.’
> > > > > > >
> > > > > > > We hope these comments from other reviewers can help facilitate resolving any remaining misunderstandings.
> > > > > > >
> > > > > > > Thank you again for your feedback. **We appreciate the time and effort the reviewer invested in this paper.** We welcome any additional questions or suggestions to further enhance the clarity of our work.

---

> > > > > > > > ### Comment · Reviewer_pW35 · 2024-12-02
> > > > > > > >
> > > > > > > > The illustration clarifies some of my questions but the inherent limitations of this version, both in presentation and in technique still need to be enhanced in the future. The federated collaboration issue is overlooked and not explained by the authors. The authors state that the assumption for perfect feature extractor is followed by LP-FT. In centralized learning, it is more feasible to obtain such model, but in federated scenario, it is seldomly resolved, and the authors simply follow the training procedure of FedAvg. Besides, the heterogeneity is not well-discussed in the previous version, the authors are supposed to extend the related work. Moreover, the reason for so many misunderstandings lies in the lack of clarity in explanation in this version, I hope this will be improved in future versions.

---

> > > > > > > > > ### Author Response · Authors · 2024-12-04
> > > > > > > > >
> > > > > > > > > We are pleased that **our response has effectively addressed the concerns raised by the reviewer**, as the two key concerns on the Non-IID nature of our setting and our theoretical distinction to a centralized setting raised by the reviewer in the last round have not been proposed. We would be happy to further clarify the following **remaining minor misunderstandings and uncertainties** by presenting our previous response **in a more accessible and concise manner**.
> > > > > > > > >
> > > > > > > > > ---
> > > > > > > > >
> > > > > > > > > > “The federated collaboration issue is overlooked and not explained by the authors. “
> > > > > > > > >
> > > > > > > > > First of all, while we understand your question on fine-tuning based on a global model trained with FedAvg, we view this **initialization** leverages the collective insights from federated training, setting a strong foundation for subsequent fine-tuning. We respectfully clarify that our **experiment, method, and theory** all account for the **unique settings in FL**. Specifically, we have comprehensively addressed the relationship between our work and federated collaboration in our above responses:
> > > > > > > > >
> > > > > > > > > - Methodology: See our detailed explanation in **Response A.5**.
> > > > > > > > > - Experiments: See **Response A.2**.
> > > > > > > > > - Theory: See **Response B.1.2**.
> > > > > > > > >
> > > > > > > > > The relevant content is **explicitly presented in our original text** (see **Section 3.1 and 3.2**) and has been further highlighted in the revised version (see **Appendix C.2 and C.3**).
> > > > > > > > >
> > > > > > > > > Second, we would like to point out that **FL research is not just about straightforward collaborative training** (see **Section 1 Line 43-47**). **The research question of what constitutes an effective personalization strategy is an important and active area of research in FL**, as evidenced by prior works (see **Section 1 Line 42-51**). Our work aims to advance this line of inquiry by
> > > > > > > > >
> > > > > > > > > - i) performing a very thorough evaluation of existing approaches,
> > > > > > > > > - ii) identifying the fundamental issues of “federated feature distortion,” and
> > > > > > > > > - iii) showing that LP-FT is a simple and easy fix.
> > > > > > > > >
> > > > > > > > > **These directly address the challenges and opportunities of personalization within the FL paradigm, and we believe they are impactful and relevant to the FL community.**
> > > > > > > > >
> > > > > > > > > ---
> > > > > > > > >
> > > > > > > > > > “The authors state that the assumption for perfect feature extractor is followed by LP-FT. In centralized learning, it is more feasible to obtain such model, but in federated scenario, it is seldomly resolved, and the authors simply follow the training procedure of FedAvg.”
> > > > > > > > >
> > > > > > > > > We believe there still remains a **misunderstanding based on the verbal interpretation of the “perfect feature extractor”**. We want to clarify that the assumption states “The pre-trained model primarily captures the shared component ($B_*$) without explicitly learning client-specific parts” This means that the global model is designed to emphasize **shared representations** across clients rather than client-specific information. In this context, we **respectfully disagree** with the reviewer’s statement that “in centralized learning, it is more feasible to obtain such a model.”  In the FL context, the feasibility of client collaboration depends on the presence of shared representations within the global data distribution, which forms **the foundation for the advantages of collaboration over independent local training**.（see detailed explanation on client collaboration in **Response A.5**)
> > > > > > > > >
> > > > > > > > > Empirically, while we follow the FedAvg procedure for pre-training, this approach is specifically aimed at extracting these shared components, forming the foundation for effective collaboration in FL. We acknowledge that empirical implementations of algorithms like FedAvg may exhibit performance gaps relative to an ideal feature extractor. However, **this does not compromise the validity of our theoretical assumption**. Moreover, we demonstrated that LP-FT is a simple yet effective tool, even when paired with the straightforward FedAvg strategy. This **further highlights the practical value** of our approach. Therefore, we respectfully disagree demonstrating the effectiveness of our proposed method based on FedAvg is a disadvantage.

---

> > > > > > > > > > ### Author Response · Authors · 2024-12-04
> > > > > > > > > >
> > > > > > > > > > > “Besides, the heterogeneity is not well-discussed in the previous version, the authors are supposed to extend the related work.”
> > > > > > > > > >
> > > > > > > > > > We are glad that the reviewer now acknowledges the heterogeneous nature of our setting. In the revised version, we have adequately extended and detailed the discussion of heterogeneity and our setting in **Section 3** and the **Appendix C**.
> > > > > > > > > >
> > > > > > > > > > We will include more relevant and advanced works on data heterogeneity in FL in the final version to further support understanding, particularly for readers who may be less familiar with the latest developments in this area.
> > > > > > > > > >
> > > > > > > > > > ---
> > > > > > > > > >
> > > > > > > > > > > “Moreover, the reason for so many misunderstandings lies in the lack of clarity in explanation in this version, I hope this will be improved in future versions.”
> > > > > > > > > >
> > > > > > > > > > We respectfully clarify that the misunderstandings mentioned are limited to two specific aspects raised by the reviewer: the “**Heterogeneity Setting**” and “**Our theoretical assumption’s distinction from the centralized setting**.” **These concerns have been thoroughly and carefully addressed in our response** (see **Response A and B.1** for the heterogeneity setting concern, and **Response B.2** for the theoretical assumption’s distinction), with detailed clarifications provided for each point.
> > > > > > > > > >
> > > > > > > > > > After extensive discussions, we are pleased that most FL domain experts among the reviewers now understand our work well. However, we recognize that the paper could be made more accessible to non-experts in FL. To enhance its readability, we will provide additional context in the final version for readers with limited familiarity with personalization and data heterogeneity in FL. We will strive to improve this aspect in the revised manuscript. Hope this clarifies it. Many thanks!
> > > > > > > > > >
> > > > > > > > > > ---
> > > > > > > > > >
> > > > > > > > > > Finally, we sincerely appreciate the time and effort invested by the reviewer. In our final version, we will address these minor issues by providing additional relevant background on federated learning heterogeneity and personalization to improve accessibility for a broader audience.
> > > > > > > > > >
> > > > > > > > > > Thank you!

---

### Official Review · Reviewer_gFiS · 2024-11-04

**Soundness:** 3
**Presentation:** 3
**Contribution:** 2
**Rating:** 6
**Confidence:** 4

**Summary:**

The authors propose an approach LP-FT to tackle the key challenge of balancing local personalization and global generalization. They conduct extensive empirical evaluation. They combined linear probing with full fine-tuning and provided a solid theoretical analysis using a two-layer linear network. Extensive experiments are conducted to evaluate the performance of the proposed mechanism.

**Strengths:**

1. The motivation and overall presentation of the paper is clear.
2. The paper is well-written.
3. The overall approach of using linear probing with full fine-tuning to prevent feature distortion is interesting.
4. The paper presents comprehensive experimental observation and theoretical analysis.
5. The proposed method shows impressive accuracy improvement on the datasets.

**Weaknesses:**

1. This work focuses on the PFL and GFL in feature distribution shift scenarios. Previous PFL work mostly works on label distribution skew. Feature-skewed FL adopts PFL paradigms. Why this work emphasizes the feature skew in personalization and generalization? What about this work’s insight on vanilla label distribution shifts?
2. The novelty of proposed method is somehow limited. It is mostly based on LP-FT. The contribution on comprehensive analysis is significant, while the novelty of technical method seems weak.
3. The organization of sections is somehow confusing. It could be better to make it clear as observations, methods, and experiments.
4. Lack of comparison with PFL baselines. The author only considers different fine-tuning paradigms.

**Questions:**

Please refer to weakness.

---

> ### Author Response · Authors · 2024-11-25
>
> **Response to W1**
>
> We would like to provide additional context and clarify our focus:
>
> **1. Importance of Feature Distribution Shifts**
> Feature distribution shifts are critical in many **real-world scenarios**, such as FL applications in healthcare and autonomous systems, where input features (e.g., sensor readings or image characteristics) vary significantly across clients due to differences in equipment, environments, or acquisition conditions (added in **Section 1 Line 77-78** in revised version). Addressing feature shifts is essential for robust personalization and generalization in such settings.
>
> **2. Limitations of PFL Methods for Label Distribution Skew**
> While many PFL methods have been developed for label distribution skew, these approaches do not necessarily generalize to feature distribution shift scenarios. As highlighted in our revised **related work** [1, 2], **some methods tailored for label skew fail to effectively mitigate the challenges posed by feature skew**, such as maintaining global generalization or handling federated feature distortion.
>
> **3. Focus on Theoretical Grounding for Feature Shift**
> Our theoretical analysis is grounded in **feature shift assumptions** to explain the phenomena studied and the effectiveness of our method. These assumptions include specific conditions on the data distribution of different clients (e.g., $\mathbb{E}_{x \sim \mathcal{D}_i}[x x^T] = I_d$ or $x_i = e_i + \epsilon n, \text{ where } n \sim \mathcal{N}(0, I)$) as well as assumptions on each client’s data-generating function (Assumption 4.1). These assumptions are considered in both concept shift and covariate shift scenarios. This framework enables us to analyze the global performance of LP-FT and FT through the lens of gradient descent under these distribution shifts. Overall, our theoretical analysis, recognized by the Reviewer **7VAu** as a strength of our work, is particularly built upon feature shift assumptions. We chose to focus on feature shifts to ensure consistency with our theoretical framework and to maintain clarity for readers, thereby avoiding potential confusion.
>
> **4. Results on Label Shifts**
> Nevertheless, we acknowledge the importance of label distribution shifts. To address your concern, we conducted additional experiments on label shifts and found that our method performs robustly in these settings as well, as seen in **Table 6 in the Appendix**.
>
> Reference:
> - [1] Guo, Yongxin, Xiaoying Tang, and Tao Lin. "FedRC: Tackling Diverse Distribution Shifts Challenge in Federated Learning by Robust Clustering." arXiv preprint arXiv:2301.12379 (2023).
> - [2] Son, Ha Min, et al. "FedUV: Uniformity and Variance for Heterogeneous Federated Learning." Proceedings of the IEEE/CVF Conference on Computer Vision and Pattern Recognition. 2024.

---

> ### Author Response · Authors · 2024-11-25
>
> **Response to W2**
>
> We would like to clarify that the novelty and contribution of our work extend beyond simply adapting an existing method (LP-FT [1]) to a new scenario (PFT). Our key contributions lie in the **methodological and theoretical novelty**, as well as **empirical findings, benchmarking, and analysis**. These efforts provide **deeper insights** and **fine-grained evaluations** of diverse distribution shifts, demonstrating the effectiveness of LP-FT in the **new PFT setting**.
>
> **Methodological Novelties (See Section 1 Contribution Insight)**
> **1. Simple yet Non-Trivial Method: Novel Focus on Final-Stage LP-FT in FL**
> While our method may appear straightforward, its novelty lies in **strategically shifting the insertion phase** of LP-FT to the **final local training stage** in FL. Unlike most existing PFL methods that require modifications to the entire training process, including both General FL (GFL) training and local personalization, our approach focuses on the **critical yet underexplored final local training stage**. This makes our method a **versatile solution**, **independent of the GFL training process**. This shift not only introduces a **fresh perspective on FL personalization** but also **demonstrates significant theoretical and empirical insights**.
>
> We also respectfully wish reviewers to agree that **Simplicity does not diminish the novelty of our method**; instead, it enhances its practical impact and potential for widespread applicability in real-world FL scenarios. A simple yet effective method, like ours, is a strength that enables broader adoption while providing substantial improvements over state-of-the-art techniques.
>
> **2. Pioneering LP-FT to FL**
> As far as we know, this work is **the first to systematically explore and leverage** the potential of the simple yet effective **LP-FT** method within the **FL** framework. This represents a significant contribution, as it opens up a new line of research exploring the application and adaptation of LP-FT in addressing the unique challenges of FL, such as data heterogeneity and distribution shifts.
>
> **Theoretical Novelties (See Section 1 Contribution Theory)**,
>
> **1. Our theoretical setting and assumption differ significantly from that of [1].**
> While [1] assumes a **centralized setting**, we focus on **personalized fine-tuning (PFT)** in FL, a fundamentally different scenario. In our case, the pre-trained model is trained on data from all clients, reflecting a global client distribution. This contrasts with the centralized setting in [1], where the pre-trained model is trained without any notion of a global client distribution.
>
> **2. Our theoretical analysis diverges significantly from that of [1].**
> In [1], the assumption is that a single ground-truth data-generating function underlies all data points. This assumption makes it challenging to study LP-FT in different distribution shift settings in PFT for FL (e.g., concept shift). In contrast, we assume that each client has a distinct ground-truth function, enabling the exploration of personalized fine-tuning in FL under varying distribution shifts. Specifically, the way we define the data-generating function (**Assumption 4.1**), with unique linear heads before pre-training $V_i^* = \begin{bmatrix} {V_{com}^*}^T & \lambda e_i^T \end{bmatrix}^T $ for each client and the way we define data heterogeneity $\Bigl( x_i = e_i + \epsilon n, \text{ with }  n \sim \mathcal{N}(0, I) \Bigr) $, allow us to study both concept shift and covariate shift using gradient descent to compare FT and LP-FT in terms of global performance. This approach provides new insights into the challenges of PFT in FL. In other words, the main challenge of our theoretical analysis lies in incorporating assumptions that align the theoretical setup with our specific framework. Our analysis connects the superior global performance of the LP-FT method to the concept of feature distortion. This novel perspective offers valuable insights into the effectiveness of LP-FT in FL through the lens of feature distortion.
>
> **3. In-depth Integrating Theoretical Analysis and Empirical Studies**
> We have conducted additional empirical studies to substantiate the theoretical analysis on the heterogeneity level $\lambda$. These include a comprehensive investigation with diverse ablation studies across varying levels of label-flipping ratios (**Section 5**). This **significantly extends the scope and granularity of distribution shifts** explored in the original LP-FT paper [1], providing deeper insights into its robustness and applicability under heterogeneous conditions.
>
> **[Continued below]**

---

> ### Author Response · Authors · 2024-11-25
>
> **[Continued from above]**
>
> **Response to W2**
>
> **Empirical Contributions (See Section 1 Contribution Evaluation)**,
>
> **1. New Observations: Detailed Analysis on Overfitting in PFT**
> As shown in **Figure 2**, we analyze commonly used fine-tuning strategies and demonstrate their **tendency to overfit**, even with carefully tuned hyperparameters, across three different types of distribution shifts. These findings highlight a critical gap in prior FL research, where such tendencies **have not been systematically explored**.
>
> **2. Comprehensive Benchmarking PFT**
> Our experiments span **seven diverse datasets** (CIFAR-10-C, CIFAR-10, Digit5, DomainNet, Chexpert, CelebA, CIFAR-10), addressing **four types of distribution shifts** (input-level shift, feature shift, spurious correlation, and label shift, as detailed in **Appendix Table 6**). We evaluate **five Personalized Fine-Tuning (PFT) methods** using **five robust metrics**, providing a thorough and comprehensive analysis.
>
> **3. Establishing New Ablation Studies to Support Theoretical Results**
> In **Figure 4**, we conduct detailed analysis on feature distortion relation to overfitting and in **Table 2**, we conduct large ranges of label flipping to support the theoretical analysis validating our theoretical insights in low heterogeneity regimes and further demonstrating LP-FT’s superior performance in scenarios with higher heterogeneity.
>
>
> Reference
> - [1] Ananya Kumar, Aditi Raghunathan, Robbie Matthew Jones, Tengyu Ma, and Percy Liang. Fine-tuning can distort pretrained features and underperform out-of-distribution. In ICLR. OpenReview.net, 2022.

---

> ### Author Response · Authors · 2024-11-25
>
> **Response to W3**
>
> Thank you for the insightful suggestion. We have revised the section structure and naming to align more clearly with observations, methods, and experiments, improving readability and coherence. We believe this refinement enhances the clarity and flow of the paper.
>
>
> ---
>
>
> **Response to W4**
>
> **1. Clarification: Our PFT setting is distinct from PFL methods.**
> irst, we would like to clarify our focus: it is essential to distinguish PFT from PFL. **PFL** focuses on **comprehensive architectural or pipeline-level modifications** to address data heterogeneity across the entire FL framework. In contrast, **our focus PFT** specifically targets the **final localized fine-tuning stage**, aiming to optimize the trade-off between local personalization and global generalization. These two approaches are **complementary but distinct**, addressing different aspects of Federated Learning. Our study centers on PFT as a modular, lightweight solution that complements broader PFL strategies without requiring significant system-level changes.
>
> **2. Clarification on the value of LP-FT research in FL**
> We acknowledge the rapid advancements in Personalized Federated Learning (PFL) research. However, studying the effectiveness of LP-FT remains valuable because it offers a **lightweight, modular solution** that integrates seamlessly into existing FL frameworks without requiring architectural changes. Unlike complex PFL methods, LP-FT is a practical, **plug-and-play** approach adaptable to diverse FL applications. **Investigating its effectiveness across various distribution shifts** provides key insights and inspires future FL research, where LP-FT can serve as a foundational module for more advanced methods.
>
> **3. Clarification: Included advanced PFL baselines.**
> We have included recent and advanced PFL baselines in **Appendix Table 4** (updated to **Appendix Table 5 in the revised version**) to provide comprehensive comparisons. However, to maintain the focus on the primary contributions of this work, these baselines are not presented in the main text. Importantly, even these state-of-the-art PFL methods exhibit **overfitting issues** and relatively low global performance compared to LP-FT. This highlights LP-FT’s **continued relevance and practical advantages**, reinforcing its role as a reliable and effective strategy for addressing data heterogeneity in Federated Learning.

---

> > ### Comment · Reviewer_gFiS · 2024-11-27
> >
> > I appreciate the authors' responses. My concerns have been addressed. I think the current positive score is fair.

---

> > > ### Author Response · Authors · 2024-11-27
> > >
> > > Thank you very much for the positive score and your constructive feedback. We will further polish the paper in the final revision. Thank you!

---

### Official Review · Reviewer_7VAu · 2024-11-04

**Soundness:** 3
**Presentation:** 4
**Contribution:** 3
**Rating:** 6
**Confidence:** 3

**Summary:**

This paper tackles the challenges of balancing personalization and global generalization in federated personalized fine-tuning. The authors perform comprehensive evaluations on various scenarios (e.g., different types of data heterogeneity, datasets, and models) to demonstrate the limitation of existing methods, where the model's generalizability will be degraded by excessive personalization. By identifying the critical cause of this overfitting problem  (i.e., feature distortion), the authors introduce LP-FT, a strategy that combines linear probing and full fine-tuning, to solve this problem. Through comprehensive empirical evaluations and theoretical analyses, the authors demonstrate that LP-FT consistently outperforms existing approaches by preserving pre-trained global features, thereby mitigating the adverse effects of .

**Strengths:**

**S1**. The manuscript is well organized in presentation and easy to understand.

**S2**. The solution to the problem has a good motivation. The authors first observe that the feature distortion before the last layer is highly correlated to the model generalizability.  Based on this observation, they introduce LP-FT to preserve global features during the fine-tuning process, thus better preserving global knowledge than existing methods.

**S3**. The proposed method is simple but effective. It effectively balances local personalization and global generalization, addressing the limitations of existing PFT methods that tend to overfit and exhibit inconsistent performance.

**S4**. Extensive experiments and theories were conducted to verify the advantages of the proposed method in preserving the generability when personalizing models.

**Weaknesses:**

**W1**. There has been a rapid evolution in personalized fine-tuning techniques in FL [1]. The latest baseline in this paper was published at two years ago (e.g., 2022). More evidences are needed to illustrate whether feature distortion is still a common problem for recent methods.

**W2**. I'm concerned about the novelty of this work. The main contribution of this work lies in adapting an existing method to a new scenario (e.g., personalized fine-tuning) to solve a widely studied problem in the past years (i.e., feature distortion [2][3]). Please further highlight the difference of this work to existing methods.

[1] Tamirisa R, Xie C, Bao W, et al. FedSelect: Personalized Federated Learning with Customized Selection of Parameters for Fine-Tuning[C]//Proceedings of the IEEE/CVF Conference on Computer Vision and Pattern Recognition. 2024: 23985-23994.

[2] Li X, Jiang M, Zhang X, et al. Fedbn: Federated learning on non-iid features via local batch normalization[J]. arXiv preprint arXiv:2102.07623, 2021.

[3] Collins L, Hassani H, Mokhtari A, et al. Exploiting shared representations for personalized federated learning[C]//International conference on machine learning. PMLR, 2021: 2089-2099.

**Questions:**

Please see the weakness.

---

> ### Author Response · Authors · 2024-11-25
>
> **Response to W1**
>
> **1. Clarification: Our PFT setting is distinct from PFL methods.**
> First, we would like to clarify our focus: it is essential to distinguish PFT from PFL, including methods like FedSelect mentioned by the reviewer. **PFL** focuses on **comprehensive architectural or pipeline-level modifications** to address data heterogeneity across the entire FL framework. In contrast, **our focus PFT** specifically targets the **final localized fine-tuning stage**, aiming to optimize the trade-off between local personalization and global generalization. These two approaches are **complementary but distinct**, addressing different aspects of Federated Learning. Our study centers on PFT as a modular, lightweight solution that complements broader PFL strategies without requiring significant system-level changes.
>
> **2. Clarification on the value of LP-FT research in FL**
> We acknowledge the rapid advancements in Personalized Federated Learning (PFL) research. However, studying the effectiveness of LP-FT remains valuable because it offers a **lightweight, modular solution** that integrates seamlessly into existing FL frameworks without requiring architectural changes. Unlike complex PFL methods, LP-FT is a practical, **plug-and-play** approach adaptable to diverse FL applications. **Investigating its effectiveness across various distribution shifts** provides key insights and inspires future FL research, where LP-FT can serve as a foundational module for more advanced methods.
>
> **3. Clarification: Included advanced PFL baselines.**
> We have included recent and advanced PFL baselines in **Appendix Table 4** (updated to **Appendix Table 5 in the revised version**) to provide comprehensive comparisons. However, to maintain the focus on the primary contributions of this work, these baselines are not presented in the main text. Importantly, even these state-of-the-art PFL methods exhibit **overfitting issues** and relatively low global performance compared to LP-FT. This highlights LP-FT’s **continued relevance and practical advantages**, reinforcing its role as a reliable and effective strategy for addressing data heterogeneity in Federated Learning.

---

> ### Author Response · Authors · 2024-11-25
>
> **Response to W2**
>
> We would like to clarify that the novelty and contribution of our work extend beyond simply adapting an existing method (LP-FT [1]) to a new scenario (PFT). Our key contributions lie in the **methodological and theoretical novelty**, as well as **empirical findings, benchmarking, and analysis**. These efforts provide **deeper insights** and **fine-grained evaluations** of diverse distribution shifts, demonstrating the effectiveness of LP-FT in the **new PFT setting**.
>
> **Methodological Novelties (See Section 1 Contribution Insight)**
>
> **1. Simple yet Non-Trivial Method: Novel Focus on Final-Stage LP-FT in FL**
> While our method may appear straightforward, its novelty lies in **strategically shifting the insertion phase** of LP-FT to the **final local training stage** in FL. Unlike most existing PFL methods that require modifications to the entire training process, including both General FL (GFL) training and local personalization, our approach focuses on the **critical yet underexplored final local training stage**. This makes our method a **versatile solution**, **independent of the GFL training process**. This shift not only introduces a **fresh perspective on FL personalization** but also **demonstrates significant theoretical and empirical insights**.
>
> We also respectfully wish reviewers to agree that **Simplicity does not diminish the novelty of our method**; instead, it enhances its practical impact and potential for widespread applicability in real-world FL scenarios. A simple yet effective method, like ours, is a strength that enables broader adoption while providing substantial improvements over state-of-the-art techniques.
>
> **2. Pioneering LP-FT to FL**
> As far as we know, this work is **the first to systematically explore and leverage** the potential of the simple yet effective **LP-FT** method within the **FL** framework. This represents a significant contribution, as it opens up a new line of research exploring the application and adaptation of LP-FT in addressing the unique challenges of FL, such as data heterogeneity and distribution shifts.
>
> **Theoretical Novelties (See Section 1 Contribution Theory)**,
>
> **1. Our theoretical setting and assumption differ significantly from that of [1].**
> While [1] assumes a **centralized setting**, we focus on **personalized fine-tuning (PFT)** in FL, a fundamentally different scenario. In our case, the pre-trained model is trained on data from all clients, reflecting a global client distribution. This contrasts with the centralized setting in [1], where the pre-trained model is trained without any notion of a global client distribution.
>
> **2. Our theoretical analysis diverges significantly from that of [1].**
> In [1], the assumption is that a single ground-truth data-generating function underlies all data points. This assumption makes it challenging to study LP-FT in different distribution shift settings in PFT for FL (e.g., concept shift). In contrast, we assume that each client has a distinct ground-truth function, enabling the exploration of personalized fine-tuning in FL under varying distribution shifts. Specifically, the way we define the data-generating function (**Assumption 4.1**), with unique linear heads before pre-training $ V_i^* = \begin{bmatrix} {V_{com}^*}^T & \lambda e_i^T \end{bmatrix}^T $ for each client and the way we define data heterogeneity $ \Bigl( x_i = e_i + \epsilon n , \text{ with }  n \sim \mathcal{N}(0, I) \Bigr)$, allow us to study both concept shift and covariate shift using gradient descent to compare FT and LP-FT in terms of global performance. This approach provides new insights into the challenges of PFT in FL. In other words, the main challenge of our theoretical analysis lies in incorporating assumptions that align the theoretical setup with our specific framework. Our analysis connects the superior global performance of the LP-FT method to the concept of feature distortion. This novel perspective offers valuable insights into the effectiveness of LP-FT in FL through the lens of feature distortion.
>
> **3. In-depth Integrating Theoretical Analysis and Empirical Studies**
> We have conducted additional empirical studies to substantiate the theoretical analysis on the heterogeneity level $\lambda$. These include a comprehensive investigation with diverse ablation studies across varying levels of label-flipping ratios (**Section 5**). This **significantly extends the scope and granularity of distribution shifts** explored in the original LP-FT paper [1], providing deeper insights into its robustness and applicability under heterogeneous conditions.
>
> **[Continued below]**

---

> ### Author Response · Authors · 2024-11-25
>
> **[Continued from above]**
>
> **Response to W2**
>
> **Empirical Contributions (See Section 1 Contribution Evaluation)**,
>
> **1. New Observations: Detailed Analysis on Overfitting in PFT**
> As shown in **Figure 2**, we analyze commonly used fine-tuning strategies and demonstrate their **tendency to overfit**, even with carefully tuned hyperparameters, across three different types of distribution shifts. These findings highlight a critical gap in prior FL research, where such tendencies **have not been systematically explored**.
>
> **2. Comprehensive Benchmarking PFT**
> Our experiments span **seven diverse datasets** (CIFAR-10-C, CIFAR-10, Digit5, DomainNet, Chexpert, CelebA, CIFAR-10), addressing **four types of distribution shifts** (input-level shift, feature shift, spurious correlation, and label shift, as detailed in **Appendix Table 6**). We evaluate **five Personalized Fine-Tuning (PFT) methods** using **five robust metrics**, providing a thorough and comprehensive analysis.
>
> **3. Establishing New Ablation Studies to Support Theoretical Results**
> In **Figure 4**, we conduct detailed analysis on feature distortion relation to overfitting and in **Table 2**, we conduct large ranges of label flipping to support the theoretical analysis validating our theoretical insights in low heterogeneity regimes and further demonstrating LP-FT’s superior performance in scenarios with higher heterogeneity.
>
>
> Reference
> - [1] Ananya Kumar, Aditi Raghunathan, Robbie Matthew Jones, Tengyu Ma, and Percy Liang. Fine-tuning can distort pretrained features and underperform out-of-distribution. In ICLR. OpenReview.net, 2022.

---

> ### Comment · Reviewer_7VAu · 2024-11-26
>
> Thank you for your detailed response; it has resolved my confusion. I will maintain my positive rating.

---

> > ### Author Response · Authors · 2024-11-26
> >
> > Thank you very much for the positive score and your constructive feedback. We will further polish the paper in the final revision. Thank you!

---

### Meta-Review · Area_Chair_H2v1 · 2024-12-15

**Metareview:**

This paper examines the challenges of adapting global models to local data in federated learning with diverse client data distributions. Through extensive evaluations, the authors identify limitations in existing personalized fine-tuning (PFT) methods, such as overfitting and inconsistent performance across distribution shifts. To address these issues, they propose LP-FT, a strategy that combines Linear Probing with full fine-tuning.

Overall, while the reviewers appreciate the paper's contribution, inherent limitations remain. In particular, there are critical issues in presentation and general technique description that require major revision.

**Additional Comments On Reviewer Discussion:**

Overall, the reviewers were uncertain about the paper's contribution, and there was no clear consensus. Unfortunately, while I was hoping for a detailed discussion with the reviewers, they were not very active, and we were unable to achieve a clear consensus.

---

### Decision · Program_Chairs · 2025-01-22

Reject